**EMBO**
*reports*

# Molecular basis for Cdk1-regulated timing of Mis18 complex assembly and CENP-A deposition

Frances Spiller[†], Bethan Medina-Pritchard[†], Maria Alba Abad[†], Martin A Wear, Oscar Molina, William C Earnshaw & A Arockia Jeyaprakash[*] (iD)

## Abstract

The centromere, a chromosomal locus that acts as a microtubule attachment site, is epigenetically specified by the enrichment of CENP-A nucleosomes. Centromere maintenance during the cell cycle requires HJURP-mediated CENP-A deposition, a process regulated by the Mis18 complex (Mis18α/Mis18β/Mis18BP1). Spatial and temporal regulation of Mis18 complex assembly is crucial for its centromere association and function. Here, we provide the molecular basis for the assembly and regulation of the Mis18 complex. We show that the N-terminal region of Mis18BP1 spanning amino acid residues 20–130 directly interacts with Mis18α/β to form the Mis18 complex. Within Mis18α/β, the Mis18α MeDiY domain can directly interact with Mis18BP1. Mis18α/β forms a hetero-hexamer with 4 Mis18α and 2 Mis18β. However, only two copies of Mis18BP1 interact with Mis18α/β to form a hetero-octameric assembly, highlighting the role of Mis18 oligomerization in limiting the number of Mis18BP1 within the Mis18 complex. Furthermore, we demonstrate the involvement of consensus Cdk1 phosphorylation sites on Mis18 complex assembly and thus provide a rationale for cell cycle-regulated timing of Mis18 assembly and CENP-A deposition.

**Keywords** Cdk1; CENP-A deposition; centromere; HJURP; Mis18 complex
**Subject Categories** Cell Cycle; Structural Biology

## Introduction

Equal and identical distribution of chromosomes to each daughter cell during cell division is essential for maintaining genome integrity. A central player regulating this process is the kinetochore, a large proteinaceous structure assembled at a specialized region of the chromosome called the centromere [1–3]. Kinetochores physically couple chromosomes with spindle microtubules to facilitate chromosome segregation. Consequently, correct kinetochore assembly and function depends on centromeres being maintained at the right place on chromosomes [3].

In most eukaryotes, the centromeric chromatin is epigenetically defined by the enrichment of nucleosomes containing the histone H3 variant CENP-A [4–7]. To maintain centromere identity, new CENP-A must be deposited in each cell cycle. The timing of CENP-A deposition varies among species (G1—humans and G2—*Schizosaccharomyces pombe*); however, the molecular mechanisms by which it is achieved share considerable similarity [3,5,8–13]. This process is initiated by the centromere targeting of the Mis18 complex (composed of Mis18α, Mis18β, and Mis18BP1) along with canonical histone chaperones, RbAp46/48 [14–16]. Centromere association of the Mis18 complex directly or indirectly makes the underlying chromatin permissive for CENP-A deposition. Mis18 proteins have also been shown to affect histone acetylation and DNA methylation at centromeres [14,17]. Mis18 centromere association subsequently allows HJURP, a CENP-A-specific chaperone, to associate with centromeres resulting in CENP-A loading [18,19]. Finally, chromatin remodelers like MgcRacGAP, RSF, Ect2, and Cdc42 have been suggested to fully stabilize the centromeric chromatin by a poorly understood maturation process [20,21]. Timely CENP-A deposition critical for centromere maintenance and function is determined by the kinase activities of Cdk1 and Plk1, which influence the centromere association of the Mis18 complex through negative regulation and positive regulation, respectively [22,23].

The Mis18 proteins contain two structurally distinct domains (Yippee/MeDiY and a C-terminal α-helix), both of which can self-oligomerize. Previously, we and others have shown that oligomerization of Mis18 proteins is required for their centromere association and function both in *S. pombe* and humans [24,25]. The human Mis18 paralogs, Mis18α and Mis18β, directly interact with Mis18BP1 and CENP-C, respectively, facilitating Mis18 complex formation and centromere association [14,26]. However, the molecular basis for the cell cycle-dependent regulation of Mis18 complex formation, and thus, CENP-A deposition has yet to be defined.

Here, we show that Mis18BP1, through its highly conserved N-terminal region comprising amino acids 20–130 (Mis18BP1$_{20-130}$), binds Mis18α/β hetero-oligomer via the Mis18α/β MeDiY hetero-dimer to form the Mis18 complex. Characterization of the oligomeric structures of the Mis18 subcomplexes revealed that the Mis18α/β complex is a hetero-hexamer with four copies of Mis18α and two

Wellcome Trust Centre for Cell Biology, Institute of Cell Biology, University of Edinburgh, Edinburgh, UK
*Corresponding author. Tel: +44 131 6507113; E-mail: jeyaprakash.arulanandam@ed.ac.uk
[†]These authors contributed equally to this work

copies of Mis18β, while the Mis18 holo-complex is a hetero-octamer with two additional copies of Mis18BP1. We identify two conserved consensus Cdk1 phosphorylation sites (T40 and S110) within Mis18BP1$_{20-130}$ and show that phospho-mimicking mutations of these residues disrupt Mis18 complex formation. This explains the molecular basis for Cdk1-mediated timing of Mis18 assembly and CENP-A deposition.

# Results and Discussion

## The N-terminal region of Mis18BP1 comprising amino acids 20–130 directly interacts with Mis18α/β to form the Mis18 complex

Mis18BP1 contains two evolutionary conserved domains, the SANTA (residues 385–472) and SANT (residues 878–927) domains (Figs 1A and EV1A). It has previously been shown that a region overlapping the SANTA domain, Mis18BP1$_{475-878}$, interacts with CENP-C while Mis18BP1$_{1-376}$ can interact with Mis18α [15,22,26]. The amino acid sequence analysis revealed the presence of a highly conserved 110 amino acid stretch within the first 130 amino acids of Mis18BP1 (residues 20–130; Mis18BP1$_{20-130}$; Fig 1A). We hypothesized that Mis18BP1$_{20-130}$ might be the minimal region that interacts with Mis18α/β. To test this, we expressed TetR-eYPF-Mis18α in HeLa 3-8 cells containing a synthetic alphoid DNA array with tetracycline operator sequences (alphoid$^{tetO}$ array) integrated in a chromosome arm [27] and analyzed the ability of Mis18α to recruit full-length Mis18BP1 (mCherry-Mis18BP1$_{fl}$) as well as two different N-terminal fragments: mCherry-Mis18BP1$_{20-130}$ and mCherry-Mis18BP1$_{336-483}$ (covering the SANTA domain) to the tethering site. Mis18α tethered to the ectopic alphoid$^{tetO}$ array recruited Mis18BP1$_{fl}$ and Mis18BP1$_{20-130}$ more robustly compared to the Mis18BP1$_{336-483}$ (Fig 1B). As expected, Mis18α tethered to the synthetic array recruited Mis18β along with Mis18BP1$_{20-130}$ (Fig EV1B). These data suggest that the Mis18BP1$_{20-130}$ is sufficient to interact with Mis18α/β to form the Mis18 complex *in vivo*.

To confirm that Mis18BP1$_{20-130}$ can directly interact with Mis18α/β *in vitro*, individually purified recombinant proteins (Mis18α/β and Mis18BP1$_{20-130}$) were analyzed using size-exclusion chromatography (SEC) before and after complex formation. While Mis18α/β and Mis18BP1$_{20-130}$ eluted at 11.0 and 15.7 ml, respectively, the complex containing Mis18α/β and Mis18BP1$_{20-130}$ eluted at 10.9 ml (Fig 1C). The elution volume of the Mis18α/Mis18β/Mis18BP1$_{20-130}$ complex is not very different from Mis18α/β, suggesting that the binding of Mis18BP1$_{20-130}$ does not significantly alter the hydrodynamic radius of the Mis18α/β. These data, together with the *in vivo* tethering assays, confirm that Mis18BP1$_{20-130}$ is sufficient to make a stable and direct interaction with Mis18α/β to form the Mis18 complex.

## The Mis18α MeDiY domain can directly interact with Mis18BP1$_{20-130}$ *in vitro*

Mis18 proteins possess a globular MeDiY domain (Mis18α$_{MeDiY}$: 77–187 and Mis18β$_{MeDiY}$: 56–183) and a C-terminal α-helical domain (Fig 2A). Our previous structure-function analysis of *S. pombe* Mis18 revealed that its centromere association and function requires

Mis18 MeDiY dimerization [24]. Stellfox *et al* [26] showed that mutations within the conserved C-X-X-C motif of the Mis18α$_{MeDiY}$ domain perturbed its ability to interact with Mis18BP1, suggesting a direct role of Mis18α for Mis18 complex formation. However, as the C-X-X-C motif mutation is expected to affect Zn binding required to stabilize the structure, it was not clear whether the inability of this mutant to interact with Mis18BP1 is a primary or secondary consequence.

To determine whether the Mis18α$_{MeDiY}$ domain is the minimal interaction surface of Mis18α/β required for Mis18BP1$_{20-130}$ binding, we purified recombinant Mis18α$_{MeDiY}$ and tested its ability to form a complex with Mis18BP1$_{20-130}$. The SEC elution volumes of Mis18α$_{MeDiY}$ on its own and in the presence of Mis18BP1$_{20-130}$ were 11.9 and 11.5 ml, respectively, demonstrating a direct interaction between Mis18α$_{MeDiY}$ and Mis18BP1$_{20-130}$ (Fig EV2A). However, due to their almost identical molecular weights (MW), Mis18α$_{MeDiY}$ (12.4 kDa) and Mis18BP1$_{20-130}$ (12.7 kDa) migrate similarly in the SDS–PAGE (Fig EV2A, bottom panel). To unambiguously validate the Mis18α$_{MeDiY}$-Mis18BP1$_{20-130}$ interaction, we purified His-GFP-Mis18α$_{MeDiY}$ and analyzed its ability to interact with Mis18BP1$_{20-130}$ in SEC (using Superdex 200 increase 10/300). While His-GFP-Mis18α$_{MeDiY}$ on its own eluted at 13.8 ml, the His-GFP-Mis18α$_{MeDiY}$/Mis18BP1$_{20-130}$ complex eluted at 13.3 ml (Fig 2B), confirming a direct interaction between Mis18α$_{MeDiY}$ and Mis18BP1$_{20-130}$.

As Mis18α$_{MeDiY}$ and Mis18β$_{MeDiY}$ share 44% sequence similarity, we tested whether the Mis18β$_{MeDiY}$ can also directly interact with Mis18BP1. Size-exclusion chromatography analysis of a sample containing Mis18β$_{MeDiY}$ and Mis18BP1$_{20-130}$ (using Superdex 75 10/300) showed that they elute at distinct elution volumes, 13.9 and 12.2 ml, respectively, demonstrating a lack of interaction (Fig 2C). This, together with a published report [26], shows that Mis18α, rather than Mis18β, mediates Mis18BP1 interaction to form the Mis18 complex.

## Mis18BP1$_{20-130}$ binds Mis18α$_{MeDiY}$/β$_{MeDiY}$ hetero-dimer more robustly than Mis18α$_{MeDiY}$

We had previously shown that Mis18α$_{MeDiY}$ prefers to hetero-dimerize with Mis18β$_{MeDiY}$ over homo-dimerizing with itself [28]. Hence, we wanted to determine whether Mis18α$_{MeDiY}$ could still bind Mis18BP1$_{20-130}$ while part of the Mis18α$_{MeDiY}$/β$_{MeDiY}$ hetero-dimer. Ni-NTA pull-down assays were performed using purified His-GFP-Mis18α$_{MeDiY}$, His-GFP-Mis18β$_{MeDiY}$, and His-GFP-Mis18α$_{MeDiY}$/Mis18β$_{MeDiY}$ with Mis18BP1$_{20-130}$ at varying salt concentrations (75–500 mM NaCl). Consistent with the SEC analysis shown in Figs 2B and C, and EV2A, when tested in isolation, only His-GFP-Mis18α$_{MeDiY}$ could bind Mis18BP1$_{20-130}$. His-GFP-Mis18β$_{MeDiY}$ failed to interact with Mis18BP1$_{20-130}$ even under a low ionic strength condition (Fig 2D). Interestingly, His-GFP-Mis18α$_{MeDiY}$/β$_{MeDiY}$ hetero-dimer interacted with Mis18BP1$_{20-130}$ (Figs 2D and EV2B). Moreover, in a high ionic strength buffer (containing 500 mM NaCl), only His-GFP-Mis18α$_{MeDiY}$/Mis18β$_{MeDiY}$ bound Mis18BP1$_{20-130}$, demonstrating that Mis18BP1 preferentially binds Mis18α$_{MeDiY}$/β$_{MeDiY}$ hetero-dimer. These data indicate that either Mis18α$_{MeDiY}$ makes additional contacts with Mis18BP1 in the presence of Mis18β$_{MeDiY}$ or Mis18β$_{MeDiY}$ strengthens Mis18α$_{MeDiY}$-Mis18BP1$_{20-130}$ binding by

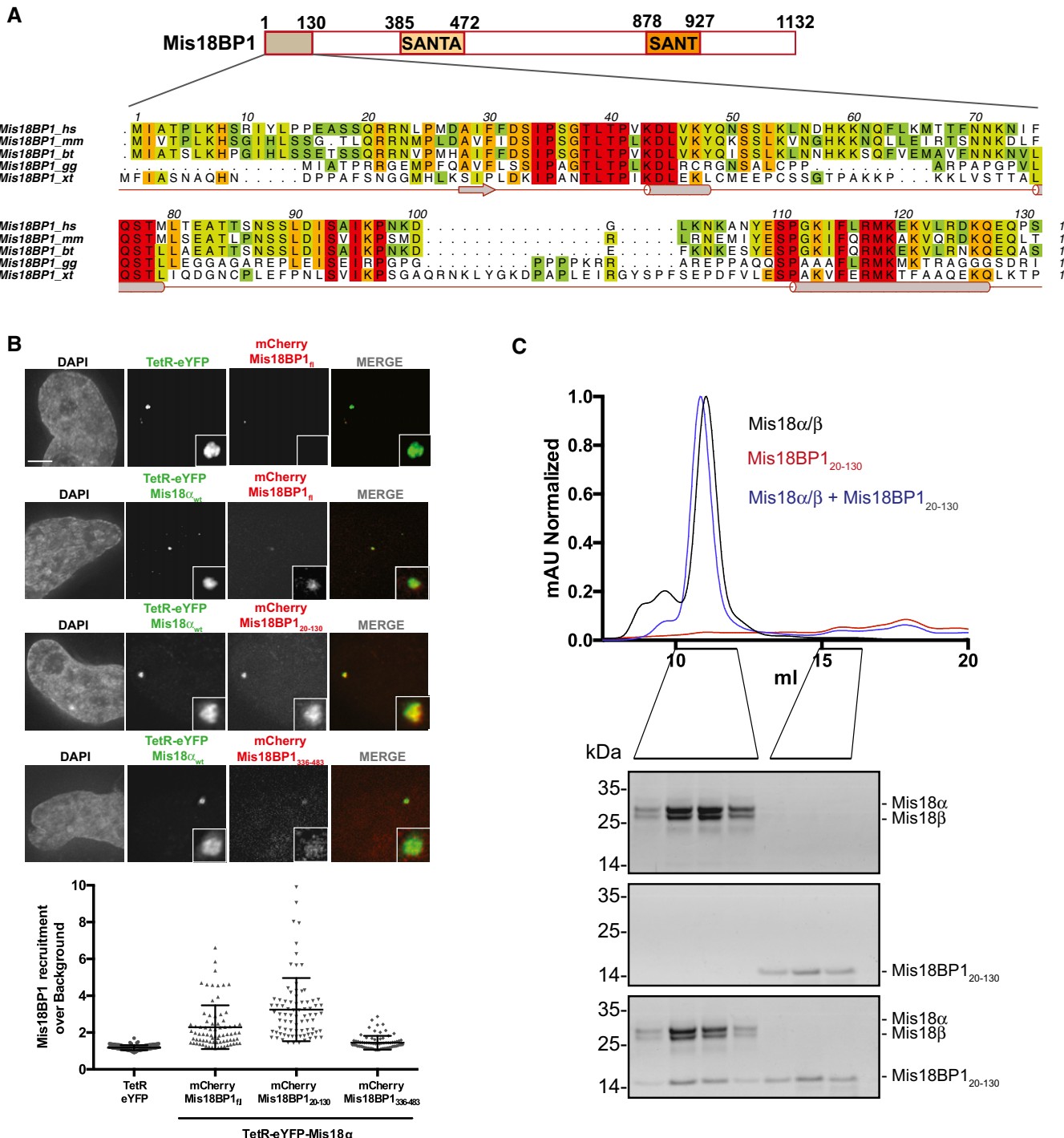

**Figure 1.  Mis18BP1$_{20-130}$ is sufficient to interact with Mis18$\alpha$/$\beta$.**

A   Schematic representation of Mis18BP1 domain architecture, amino acid conservation (conservation score is mapped from red to cyan, where red corresponds to highly conserved and cyan to poorly conserved), and secondary structure (Conserved Domain Database [CDD] and PsiPred, http://bioinf.cs.ucl.ac.uk/psipred). Alignments include *Homo sapiens* (*hs*), *Mus musculus* (*mm*), *Bos taurus* (*bt*), *Gallus gallus* (*gg*), and *Xenopus tropicalis* (*xt*). Multiple sequence alignments were performed with MUSCLE [31] and edited with Aline [32].

B   Representative fluorescence images (top) and quantification (bottom) for the analysis of mCherry-Mis18BP1 recruitment to the alphoid$^{tetO}$ array by TetR-eYFP-Mis18$\alpha$. HeLa 3-8 cells co-transfected with either TetR-eYFP or TetR-eYFP-Mis18$\alpha_{wt}$ and mCherry vectors containing the indicated versions of Mis18BP1. Middle bars show median whilst error bars show SEM. Mann–Whitney test vs. TetR-eYFP; $P \leq 0.0001$, $n \geq 77$. Scale bar, 5 µm.

C   SEC profiles and respective SDS–PAGE analysis of Mis18$\alpha$/$\beta$, Mis18BP1$_{20-130}$, and Mis18$\alpha$/$\beta$ mixed with molar excess of Mis18BP1$_{20-130}$.

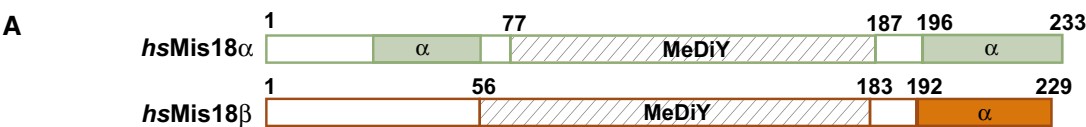

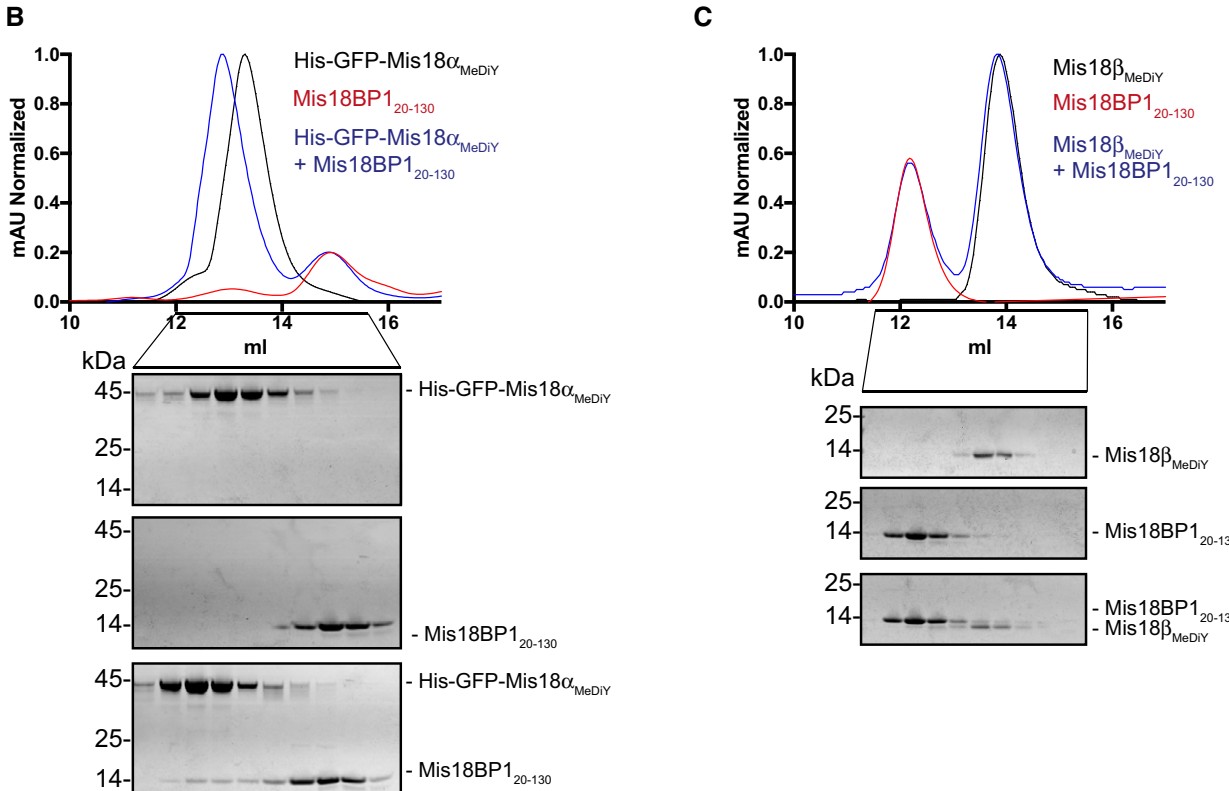

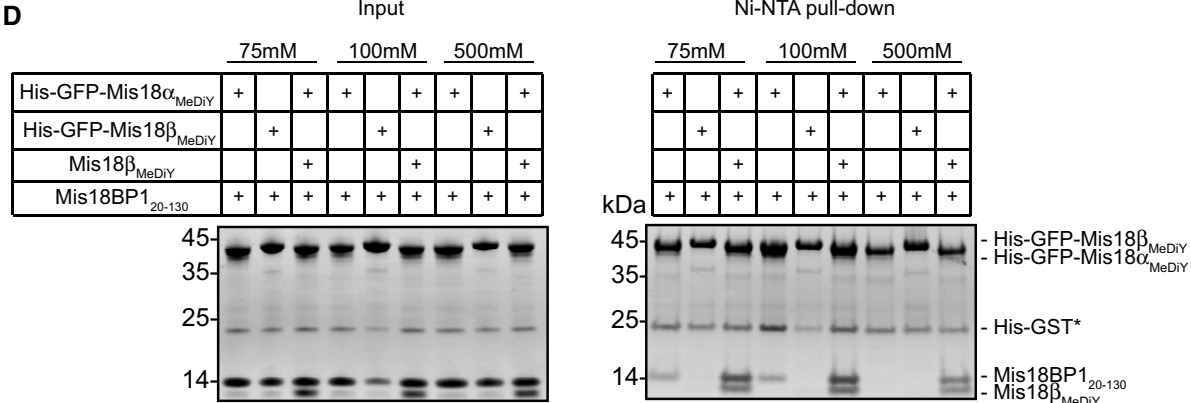

**Figure 2.  Mis18α_MeDiY domain directly interacts with Mis18BP1_20–130.**

A   Schematic representation of *hs*Mis18α, *hs*Mis18β domain architecture (using CDD and PsiPred).

B, C   SEC profiles and respective SDS–PAGE analysis of (B) His-GFP-Mis18α_MeDiY, Mis18BP1_20–130, and His-GFP-Mis18α_MeDiY mixed with molar excess of Mis18BP1_20–130 and (C) Mis18β_MeDiY, Mis18BP1_20–130, and Mis18β_MeDiY with molar excess of Mis18BP1_20–130.

D   SDS–PAGE analysis of the Ni-NTA pull-down assay where recombinant His-GFP-Mis18α_MeDiY, His-GFP-Mis18β_MeDiY, and Mis18β_MeDiY were mixed with Mis18BP1_20–130 in different combinations in either 75, 100, or 500 mM NaCl. Left panel: inputs and right panel: Ni-NTA pull-down. * Contamination carried over from Mis18BP1 purification.

directly interacting with Mis18BP1$_{20-130}$ in a Mis18$\alpha_{MeDiY}$-dependent manner.

## Mis18$\alpha$/$\beta$ forms a hetero-hexamer comprising four copies of Mis18$\alpha$ and two copies of Mis18$\beta$

We and others have shown that homo- and hetero-oligomerization of Mis18 proteins are crucial for Mis18 function in *S. pombe* [24] and in humans [25], respectively. However, the precise subunit composition of human Mis18$\alpha$/$\beta$ hetero-oligomer and its consequence on Mis18 complex assembly and function remain elusive.

First, we determined the absolute molecular mass of untagged Mis18$\alpha$/$\beta$ complex using SEC combined with multi-angle light scattering (SEC-MALS). The measured MW of the untagged Mis18$\alpha$/$\beta$ complex was 151.2 ± 2.9 kDa (calculated MWs of Mis18$\alpha$ and Mis18$\beta$ are 25.9 kDa and 24.7 kDa, respectively) and correlated well with the calculated MW of a hetero-hexamer (151.8 kDa, using the average MW of Mis18 proteins 25.3 kDa; Figs 3A and EV3A). This contrasts with a previous report by Nardi *et al* [25], which used glycerol-based gradient experiments to show that Mis18$\alpha$/$\beta$ is a hetero-tetramer.

As Mis18$\alpha$ and Mis18$\beta$ are very similar in size (~25 kDa), it is almost impossible to accurately determine the subunit stoichiometry within the Mis18 hetero-hexamer. Hence, we introduced a noticeable size variation by purifying His-GFP-Mis18$\alpha$ in a complex with His-Mis18$\beta$. Interestingly, the measured MW of His-GFP-Mis18$\alpha$/His-Mis18$\beta$ complex was 287.2 ± 5.5 kDa, matching the calculated MW of a hetero-hexamer containing four copies of His-GFP-Mis18$\alpha$ and two copies of His-Mis18$\beta$ (276 kDa; Figs 3B and EV3B). The measured MW of Mis18$\alpha$/$\beta$ complex formed using His-Mis18$\alpha$ and His-GFP-Mis18$\beta$ revealed same stoichiometry as above (measured MW = 232.2 ± 4.5 kDa; calculated MW of 4 His-Mis18$\alpha$ : 2 His-GFP-Mis18$\beta$ = 221.4 kDa; Figs 3C and EV3C) and demonstrates that Mis18$\alpha$/$\beta$ is a hetero-hexamer with a 4:2 stoichiometry.

## Mis18$\alpha$/$\beta$ hetero-hexamer is assembled from hetero-trimers of C-terminal $\alpha$-helical domains and hetero-dimers of MeDiY domains

We had previously shown that the MeDiY domains of Mis18$\alpha$ and Mis18$\beta$ form a homo-dimer and a monomer, respectively, but can form a hetero-dimer [28]. In addition, Nardi *et al* [25] have shown that the C-terminal $\alpha$-helical domains also have the ability to oligomerize. These observations together with our data that the Mis18$\alpha$/$\beta$ complex is a hetero-hexamer (Fig 3A–C) prompted us to define the stoichiometry of the Mis18$\alpha$/$\beta$ C-terminal helical assembly. Using individually purified His-GFP-Mis18$\alpha_{C\text{-term}}$ (Mis18$\alpha$ 188-end) and His-MBP-Mis18$\beta_{C\text{-term}}$ (Mis18$\beta$ 184-end), we reconstituted the C-terminal helical assembly and analyzed their composition using SEC-MALS (Figs 2A and 3D). The measured MW of His-GFP-Mis18$\alpha_{C\text{-term}}$/His-MBP-Mis18$\beta_{C\text{-term}}$ complex was 115.5 ± 2.2 kDa, which matches a calculated MW of a hetero-trimeric assembly with 2 His-GFP-Mis18$\alpha_{C\text{-term}}$ and 1 His-MBP-Mis18$\beta_{C\text{-term}}$ (119.6 kDa; Figs 3D and EV3D). This suggests that the formation of the full-length hetero-hexameric Mis18$\alpha$/$\beta$ assembly requires further oligomerization of Mis18$\alpha$/$\beta$ hetero-trimers mediated by Mis18$\alpha_{MeDiY}$/$\beta_{MeDiY}$ hetero-dimers (Fig 3E).

## The Mis18 complex is a hetero-octamer with two copies of Mis18BP1

The measured MW of the Mis18 holo-complex (Mis18$\alpha$/Mis18$\beta$/Mis18BP1$_{20-130}$) by SEC-MALS (175.4 ± 3.3 kDa) is in agreement with the calculated MW of a hetero-octamer containing a Mis18$\alpha$/$\beta$ hetero-hexamer plus two copies of Mis18BP1$_{20-130}$ (178.2 kDa; Figs 3F and EV3E). Since the MW of Mis18BP1$_{20-130}$ is only 12.7 kDa, we reconstituted the Mis18 holo-complex with His-SUMO-Mis18BP1$_{20-130}$. The measured MW of this complex (199.4 ± 3.8 kDa) matches the calculated MW (206.6 kDa; Figs 3G and EV3F) of a Mis18$\alpha$/$\beta$ hexamer bound to two copies of His-SUMO-Mis18BP1$_{20-130}$, confirming that the Mis18 holo-complex is a hetero-octamer with just two copies of Mis18BP1.

Although both Mis18$\alpha_{MeDiY}$ and Mis18$\alpha_{MeDiY}$/$\beta_{MeDiY}$ hetero-dimer can interact with Mis18BP1 (Figs 2B and D, and EV2), only two copies of Mis18BP1 bind to a Mis18$\alpha$/$\beta$ hexamer containing 2 Mis18$\alpha_{MeDiY}$ and 2 Mis18$\alpha_{MeDiY}$/$\beta_{MeDiY}$ hetero-dimers (Fig 3F and G). We propose that Mis18$\alpha_{MeDiY}$/$\beta_{MeDiY}$ hetero-dimer is preferred over Mis18$\alpha_{MeDiY}$ due its relatively stronger binding to Mis18BP1 (Figs 2D and 4A), thus limiting the number of Mis18BP1 binding sites to 2 in the holo-complex. It is also possible that the Mis18BP1 binding sites of 2 Mis18$\alpha_{MeDiY}$ domains that are not part of the Mis18$\alpha_{MeDiY}$/$\beta_{MeDiY}$ hetero-dimer are occluded in the context of the full-length Mis18$\alpha$/$\beta$ hetero-hexamer (Fig 4A).

## MeDiY dimerization interface of Mis18$\alpha$ is required for Mis18 oligomerization and Mis18BP1 binding

Our previous structural analysis of the *S. pombe* MeDiY domain identified a putative substrate-binding pocket and a dimerization interface, both required for Mis18 function [24]. To evaluate whether equivalent surfaces in human Mis18$\alpha$ are required for Mis18BP1 binding, we introduced mutations V82E/Y176D (Mis18$\alpha_{DimerM}$) at the MeDiY dimeric interface of Mis18$\alpha$/$\beta$ and L136A/Y152A/C167A/S169K (Mis18$\alpha_{PocketM}$) in the putative substrate-binding pocket using a *S. pombe*-based homology-modeled structure (Fig 4B). We then analyzed Mis18BP1 binding to His-GFP-Mis18$\alpha$/His-Mis18$\beta$ (Mis18$\alpha_{wt}$/$\beta$), His-GFP-Mis18$\alpha_{DimerM}$/His-Mis18$\beta$ (Mis18$\alpha_{DimerM}$/$\beta$), and His-GFP-Mis18$\alpha_{PocketM}$/His-Mis18$\beta$ (Mis18$\alpha_{PocketM}$/$\beta$) complexes using a Ni-NTA pull-down assay. While Mis18$\alpha_{wt}$/$\beta$ and Mis18$\alpha_{PocketM}$/$\beta$ interacted with Mis18BP1$_{20-130}$, Mis18$\alpha_{DimerM}$/$\beta$ failed to do so efficiently (Fig 4C). This suggests that the Mis18$\alpha_{MeDiY}$ interface proposed to mediate dimerization with Mis18$\beta_{MeDiY}$ is also required for Mis18BP1 binding (in agreement with Fig 2D). Size-exclusion chromatography analysis of Mis18$\alpha_{DimerM}$/$\beta$, unlike Mis18$\alpha_{wt}$/$\beta$, showed the presence of several distinct populations including an aggregated and a smaller MW species (Fig EV4). The measured MW of the smaller MW species (the sample eluting at 11.7 ml) using SEC-MALS was 129.3 ± 2.5 kDa (Figs 4D and EV4). This correlates with the calculated MW of a complex containing 2 His-GFP-Mis18$\alpha_{DimerM}$ and 1 His-Mis18$\beta$ (137.8 kDa). This strengthens the notion that the Mis18$\alpha$/$\beta$ hetero-trimer is the core oligomeric unit formed through the interactions of the C-terminal helices which then assemble into a hetero-hexamer via the MeDiY-dimerization interface (Figs 3D

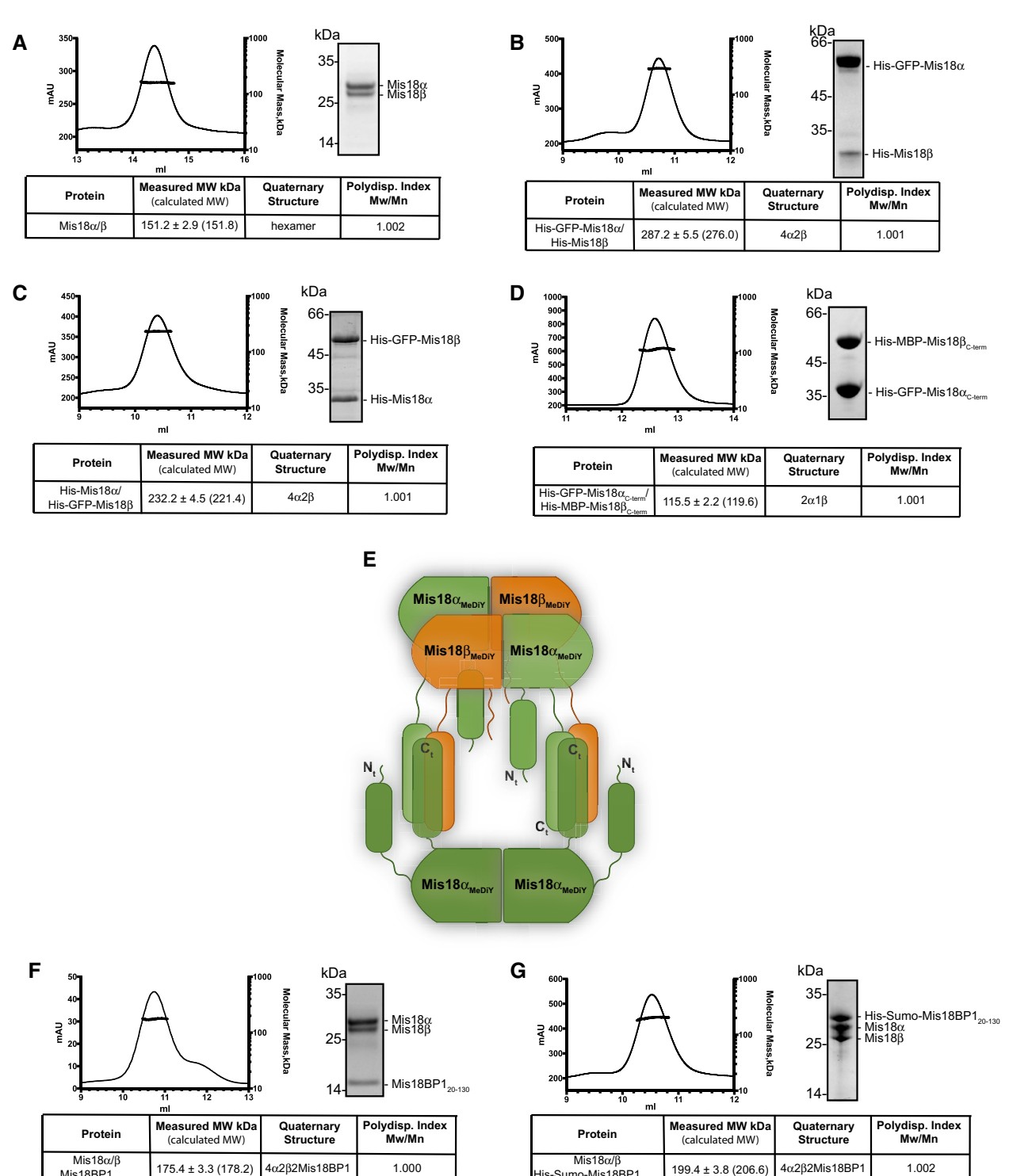

**Figure 3.  Mis18α/β is a hetero-hexamer (4 Mis18α : 2 Mis18β) which binds two copies of Mis18BP1 to form a hetero-octameric Mis18 complex.**

A–D  SEC-MALS analysis of (A) Mis18α/β complex, (B) His-GFP-Mis18α/His-Mis18β, (C) His-Mis18α/His-GFP-Mis18β, (D) His-GFP-Mis18α$_{C-term}$/His-MBP-Mis18β$_{C-term}$. Absorption at 280 nm (mAU, left *y*-axis) and molecular mass (kDa, right *y*-axis) are plotted against elution volume (ml, *x*-axis). Measured molecular weight (MW) and the calculated subunit stoichiometry based on the predicted MW of different subunit compositions (Fig EV3) are shown in the tables below.

E  Proposed model for Mis18α/β complex assembly. The Mis18α/β hetero-hexamer contains 4 Mis18α and 2 Mis18β. The C-terminal α-helical domains form a hetero-trimer (2 Mis18α and 1 Mis18β). While 2 Mis18α$_{MeDiY}$ domains engage with 2 Mis18β$_{MeDiY}$ domains to facilitate Mis18α/β hetero-hexamerization, the remaining 2 Mis18α$_{MeDiY}$ self-oligomerize. In this model, within each C-terminal α-helical assembly, Mis18α helices are shown in anti-parallel arrangement. In the absence of any structural data, an alternate model with C-terminal helices arranged in parallel orientation is equally probable.

F, G  SEC-MALS analysis of (F) Mis18α/Mis18β/Mis18BP1$_{20–130}$ and (G) Mis18α/Mis18β/His-Sumo-Mis18BP1$_{20–130}$.

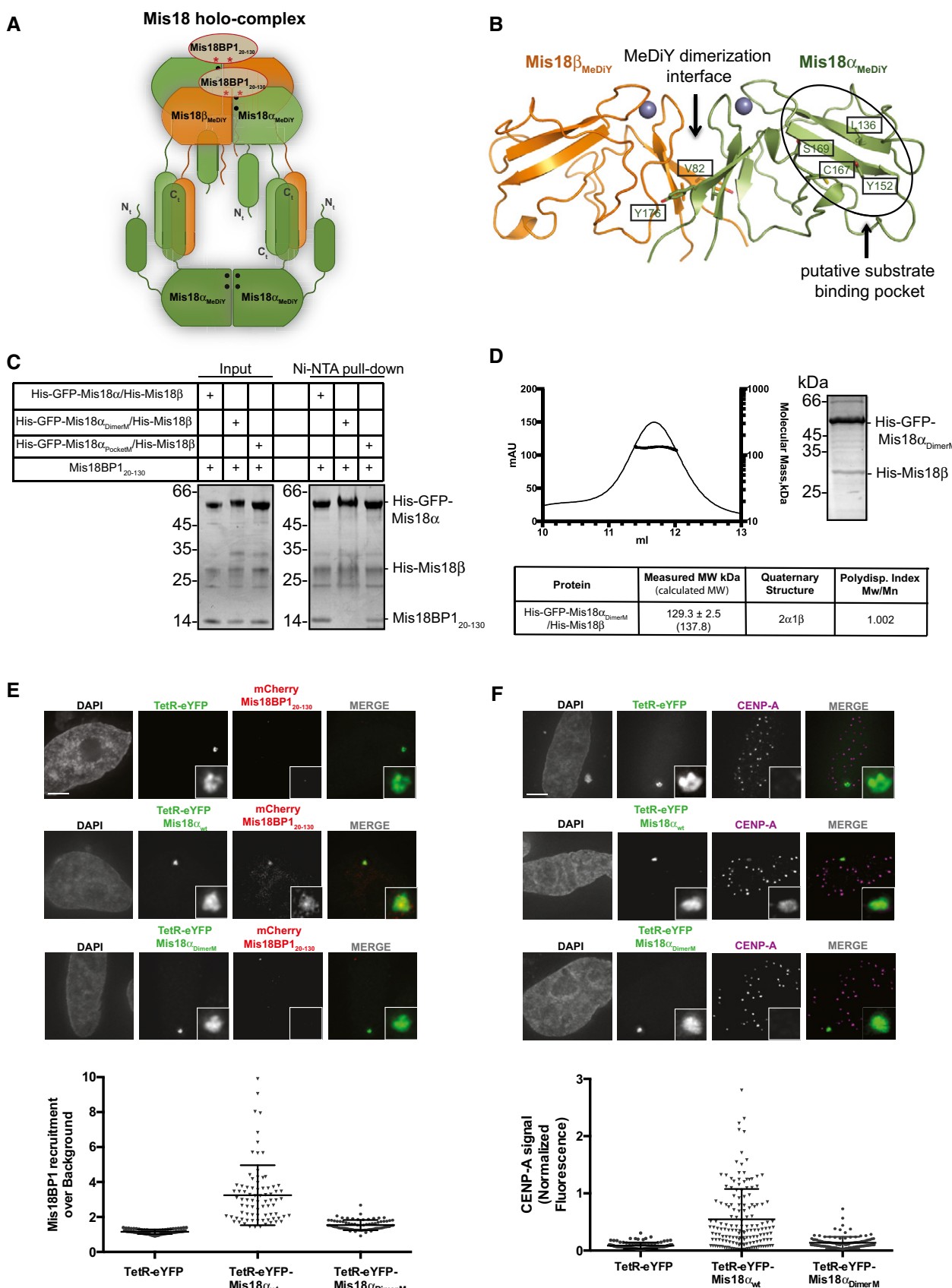

Figure 4.

◄

**Figure 4.  Hetero-hexamerization of Mis18α/β mediated via the MeDiY domain is required for Mis18BP1 binding and CENP-A loading *in vivo*.**

A     Proposed model for Mis18 complex assembly. Our data showing that the Mis18 holo-complex contains just two copies of Mis18BP1 (Fig 3F and G) and that Mis18α$_{MeDiY}$/β$_{MeDiY}$ hetero-dimer binds Mis18BP1$_{20-130}$ more robustly as compared to Mis18α$_{MeDiY}$ (Fig 2D) suggest that Mis18BP1 prefers Mis18α$_{MeDiY}$/β$_{MeDiY}$ hetero-dimer over Mis18α$_{MeDiY}$. Filled circles (•) and asterisks (*) represent the sites of Mis18α residues involved in MeDiY dimerization (V82 and Y176; Fig 4B) and consensus Cdk1 phosphorylation on Mis18BP1$_{20-130}$ (T40 and S110; Fig 5A), respectively.

B     Cartoon representation of the homology-modeled human Mis18α$_{MeDiY}$/β$_{MeDiY}$ hetero-dimer (with Phyre2 web server, www.sbg.bio.ic.ac.uk/phyre2/, using *S. pombe* Mis18 MeDiY domain—PDB: 5HJ0). Key amino acid residues forming the dimeric interface and putative substrate-binding pocket are shown in stick representation. Residues mutated in this study are highlighted by boxes.

C     SDS–PAGE analysis of the Ni-NTA pull-down assay where recombinant His-GFP-Mis18α$_{wt}$/His-Mis18β, His-GFP-Mis18α$_{PocketM}$/His-Mis18β, and His-GFP-Mis18α$_{DimerM}$/His-Mis18β were mixed with Mis18BP1$_{20-130}$. Left panel: inputs and right panel: Ni-NTA pull-down.

D     SEC-MALS profile of His-GFP-Mis18α$_{DimerM}$/His-Mis18β. Absorption at 280 nm (mAU, left *y*-axis) and molecular mass (kDa, right *y*-axis) plotted against elution volume (ml, *x*-axis).

E, F   Representative fluorescence images (top) and quantification (bottom) for the ability of TetR-eYFP, TetR-eYFP-Mis18α$_{wt}$, and TetR-eYFP-Mis18α$_{DimerM}$ (E) to recruit mCherry-Mis18BP1$_{20-130}$ to the alphoid$^{tetO}$ array (Mann–Whitney test vs. TetR-eYFP; $P \leq 0.0001$, $n \geq 78$) and (F) to deposit CENP-A at the tethering site (Mann–Whitney test vs. TetR-eYFP; $P \leq 0.0001$, $n \geq 106$). Middle bars show median whilst error bars show SEM. Scale bars, 5 μm. Data for TetR-eYFP-Mis18α$_{wt}$/mCherry-Mis18BP1$_{20-130}$-transfected cells (Fig 1B) have been used as a control.

and 4A). We conclude that the Mis18α$_{MeDiY}$ dimerization interface is required both for Mis18BP1 binding and for the higher order oligomerization of the Mis18α/β hetero-trimer (Fig 4A).

We next evaluated the contribution of Mis18α$_{MeDiY}$ dimerization interface on Mis18BP1 binding *in vivo* by tethering TetR-eYFP-Mis18α$_{DimerM}$ to the alphoid$^{tetO}$ array and probing the recruitment of mCherry-Mis18BP1$_{20-130}$. In agreement with the *in vitro* data, the TetR-eYFP-Mis18α$_{DimerM}$ failed to recruit mCherry-Mis18BP1$_{20-130}$ (Fig 4E). Furthermore, this mutant, unlike the TetR-eYFP-Mis18α$_{wt}$, was unable to deposit CENP-A at the tethering site (Fig 4F). This confirms that the Mis18α$_{MeDiY}$ dimerization interface is required for Mis18BP1 binding and CENP-A deposition.

### Cdk1 consensus sites of Mis18BP1 are directly involved in Mis18α/β binding

Mis18 complex assembly and its centromere localization have been suggested to be regulated by Cdk1 in a cell cycle-dependent manner [22,23]. Interestingly, the amino acid sequence analysis revealed two consensus Cdk1 phosphorylation sites, T40 and S110, within Mis18BP1$_{20-130}$ (Fig 5A), which we show here is able to interact with Mis18α/β (Fig 1). Notably, Mis18BP1 S110 has previously been shown to be phosphorylated *in vivo* [23]. We hypothesized that phosphorylation of Mis18BP1 T40 or/and S110 negatively regulates its association with Mis18α/β and hence Mis18 complex assembly.

We evaluated the effect of Mis18BP1 phospho-mimicking mutations T40E and S110D, either alone (Mis18BP1$_{20-130\ T40E}$ and Mis18BP1$_{20-130\ S110D}$) or in combination (Mis18BP1$_{20-130\ T40E/S110D}$), on Mis18 complex formation using SEC. Both Mis18BP1$_{20-130\ T40E}$ and Mis18BP1$_{20-130\ S110D}$ interacted with Mis18α/β. However, Mis18BP1$_{20-130\ T40E/S110D}$ failed to form a complex with Mis18α/β (Fig 5B), demonstrating a direct role of conserved Cdk1 phosphorylation sites for Mis18 complex formation *in vitro*.

Consistent with the *in vitro* data, Mis18BP1$_{20-130\ T40E}$ and Mis18BP1$_{20-130\ S110D}$ were recruited to the alphoid$^{tetO}$ array by TetR-eYFP-Mis18α, albeit less efficiently as compared to Mis18BP1$_{20-130\ wt}$ (Fig 5C). The recruitment of Mis18BP1$_{20-130\ T40E/S110D}$ mutant by TetR-eYFP-Mis18α was almost completely abolished, highlighting the involvement of the consensus Cdk1 phosphorylation sites of Mis18BP1, T40 and S110, in forming a direct Mis18α/β binding interface.

To further study the cell cycle regulation of Mis18BP1, we tested the recruitment of mCherry-Mis18BP1$_{20-130}$ to the alphoid$^{tetO}$ array by TetR-eYFP-Mis18α during different stages of mitosis (Fig 5D). Interestingly, in agreement with a previously reported suggestion [22], Mis18α was unable to recruit Mis18BP1$_{20-130\ wt}$ during early stages of mitosis when Cdk1 levels are high. On the contrary, a non-phosphorylatable mutant of Mis18BP1 (mCherry-Mis18BP1$_{20-130\ T40A/S110A}$) was recruited by TetR-eYFP-Mis18α to the array throughout the cell cycle. Consistent with this, recombinantly purified Mis18BP1$_{20-130\ T40A/S110A}$ interacted with Mis18α/β as analyzed by SEC (Fig EV5).

The temporal regulation of Mis18 complex formation defines the timing of HJURP-mediated CENP-A deposition essential for centromere inheritance and function [18,22,23]. Oligomerization of Mis18 proteins and Cdk1 activity are both key regulators of Mis18 complex assembly [22–25]. Our work presented here, in agreement with a parallel study by Pan *et al* [29], shows that the Mis18α/β complex is a hetero-hexamer made of 2 Mis18α/β hetero-trimers, each with 2 Mis18α and 1 Mis18β that are held together by the hetero-trimeric C-terminal α-helical assembly. One Mis18α$_{MeDiY}$ from each Mis18α/β trimer hetero-dimerizes with Mis18β$_{MeDiY}$ to form a Mis18α/β hetero-hexamer. This arrangement results in 2 Mis18α$_{MeDiY}$/β$_{MeDiY}$ hetero-dimers that bind two copies of Mis18BP1 and 2 Mis18α$_{MeDiY}$ that possibly form a homo-dimer (Fig 4A). However, our data show that Mis18α$_{MeDiY}$ can bind Mis18BP1, albeit less efficiently compared to Mis18α$_{MeDiY}$/β$_{MeDiY}$ hetero-dimer contradicting the suggestion by Pan *et al* [29] that neither Mis18α$_{MeDiY}$ nor Mis18β$_{MeDiY}$ can interact with Mis18BP1 on their own. The apparent contradiction could be due to the higher ionic strength buffer used by Pan *et al* [29] in their binding assays. Whether Mis18α/β hetero-hexamer can bind more than two copies of Mis18BP1 mediated via the free Mis18α$_{MeDiY}$ under specific circumstances is yet to be determined. Finally, we identified two highly conserved Cdk1 consensus sites 70 amino acids apart within the Mis18 binding region of Mis18BP1 (T40 and S110). While mutating both these amino acid residues to phospho-mimicking residues completely abolished the ability of Mis18BP1 to bind Mis18α/β *in vitro* and *in vivo*, individual phospho-mimic mutants (T40E or S110D) failed do so efficiently. This suggests that Mis18BP1 binds Mis18α/β possibly via a bipartite binding mode. Overall, our findings together with the recent independent study by Pan *et al* [29] provide key insights into the molecular basis for cell cycle-dependent Mis18 complex assembly and function.

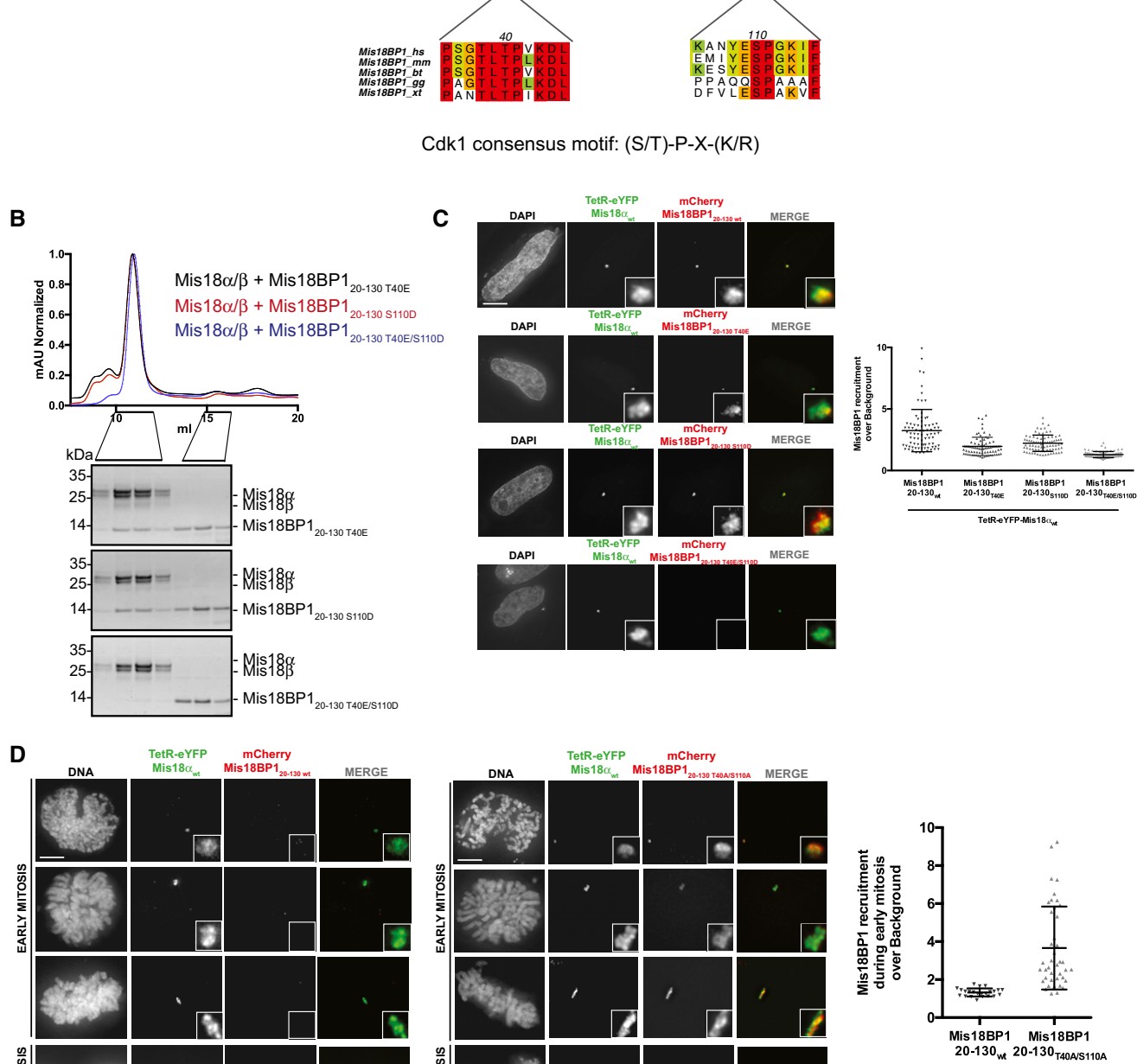

**Figure 5.** Consensus Cdk1 phosphorylation sites of Mis18BP1, T40, and S110, are directly involved in Mis18α/β binding.

A   Multiple sequence alignment of amino acid stretches containing the consensus Cdk1 phosphorylation sites within Mis18BP1$_{20-130}$. Alignments include *Homo sapiens* (hs), *Mus musculus* (mm), *Bos taurus* (bt), *Gallus gallus* (gg), and *Xenopus tropicalis* (xt). Multiple sequence alignment (conservation score is mapped from red to cyan, where red corresponds to highly conserved and cyan to poorly conserved) was performed with MUSCLE [31] and edited with Aline [32].

B   SEC profiles and respective SDS–PAGE analysis of Mis18α/β mixed with molar excess of Cdk1 phospho-mimic mutants, Mis18BP1$_{20-130\ T40E}$, Mis18BP1$_{20-130\ S110D}$, and Mis18BP1$_{20-130\ T40E/S110D}$.

C   Representative images (left) and quantification (right) for the recruitment of different mCherry-Mis18BP1$_{20-130}$ Cdk1 phospho-mimic mutants (T40E, S110D or T40E/S110D) to the alphoid$^{tetO}$ array in HeLa 3-8 cells expressing TetR-eYFP-Mis18α. Middle bars show median whilst error bars show SEM. Mann–Whitney test vs. TetR-eYFP; $P \leq 0.0001$, $n \geq 78$. Scale bar, 10 μm. Data for TetR-eYFP-Mis18α$_{wt}$/mCherry-Mis18BP1$_{20-130}$-transfected cells (Fig 1B) have been used as a control.

D   Representative images (left) for the cell cycle-dependent and independent recruitment of mCherry-Mis18BP1$_{20-130\ wt}$ and mCherry-Mis18BP1$_{20-130\ T40A/S110A}$, respectively, to the alphoid$^{tetO}$ array in HeLa 3-8 cells expressing TetR-eYFP-Mis18α. Quantification (right) of Mis18BP1$_{20-130\ wt}$ and Mis18BP1$_{20-130\ T40A/S110A}$ recruitment to the array during early mitosis. Middle bars show median whilst error bars show SEM. Mann–Whitney test vs. Mis18BP1$_{20-130\ wt}$; $P \leq 0.0001$, $n = 23$ (mCherry-Mis18BP1$_{20-130\ wt}$) and $n = 41$ (mCherry-Mis18BP1$_{20-130\ T40A/S110A}$). Scale bar, 10 μm.

# Materials and Methods

### Expression and purification of recombinant human proteins

Human Mis18α, Mis18α$_{MeDiY}$, Mis18α$_{C\text{-term}}$, Mis18β, Mis18β$_{MeDiY}$, and Mis18β$_{C\text{-term}}$ were amplified from codon-optimized sequences (GeneArt) while Mis18BP1$_{20–130}$ was amplified from a human cDNA library (MegaMan human transcriptome library, Agilent). Amplifications were then cloned into pET His6 msfGFP TEV, pET His6 TEV, pET His6 Sumo TEV, pET His6 MBP TEV, pGEX-6P-1 (GE Healthcare), pEC-K-3C-His-GST, and pEC-K-3C-His LIC vectors. pET His6 msfGFP TEV (9GFP Addgene plasmid # 48287), pET His6 TEV (9B Addgene plasmid # 48284), pET His6 Sumo TEV (14S Addgene plasmid # 48291), and pET His6 MBP TEV (9C Addgene Plasmid #48286) were a gift from Scott Gradia.

Mis18α MeDiY dimer-disrupting mutant (Mis18α$_{DimerM}$; V82E/Y176D) and putative substrate-binding pocket mutant (Mis18α$_{PocketM}$; L136A/Y152A/C167A/S169K) and Mis18BP1 mutants, Mis18BP1$_{20–130\ T40E}$, Mis18BP1$_{20–130\ S110D}$, Mis18BP1$_{20–130\ T40E/S110D}$, and Mis18BP1$_{20–130\ T40A/S110A}$ were generated following the Quikchange site-directed mutagenesis protocol (Stratagene).

Proteins were expressed, either individually or together with their binding partners, using *E. coli* BL21 gold grown in LB media. Cultures were induced with 0.35 mM IPTG at 18°C overnight. Cell lysis was carried out by sonicating cells re-suspended in a lysis buffer containing 20 mM Tris (pH 8.0), 250 mM NaCl (or 500 mM for Mis18BP1$_{20–130}$), 35 mM imidazole, and 2 mM βME. Lysis buffer was supplemented with 10 μg/ml DNase, 1 mM PMSF, and cOmplete (EDTA-free, Roche). Proteins were purified from the clarified lysates by affinity chromatography using a 5 ml HisTrap HP column (GE Healthcare). The protein-bound resin was washed with lysis buffer, followed by a buffer containing 20 mM Tris (pH 8.0), 1 M NaCl, 50 mM KCl, 10 mM MgCl$_2$, 2 mM ATP, 35 mM imidazole, 2 mM βME, and with a final lysis buffer wash. Proteins were eluted using a lysis buffer supplemented with 500 mM imidazole and dialyzed overnight into 20 mM Tris (pH 8.0), 75–100 mM NaCl, and 2 mM DTT. All proteins were subjected to anion exchange chromatography using the HiTrap Q column (GE Healthcare). Appropriate fractions were pooled, concentrated, and injected into a Superdex 200 increase 10/300 or Superdex 75 10/300 column (GE Healthcare) equilibrated with 20 mM Tris (pH 8.0), 100–250 mM NaCl, and 2 mM DTT. Fractions were analyzed on SDS–PAGE and stained with Coomassie blue.

Ni-NTA pull-down assay was performed in 20 mM Tris (pH 8.0), 75–500 mM NaCl, 10% glycerol, 0.5% NP-40, 35 mM imidazole, and 2 mM βME. Proteins were mixed with two times molar excess of Mis18BP1$_{20–130}$ and made up to 200 μl with buffer before incubated for 30 min at 4°C with 60–120 μl of slurry that had been equilibrated in buffer. Beads were then washed four times with 1 ml of buffer, and bound protein was eluted by boiling in SDS–PAGE loading dye before being analyzed by SDS–PAGE.

### SEC-MALS

Size-exclusion chromatography (ÄKTAMicro™, GE Healthcare) coupled to UV, static light scattering, and refractive index detection (Viscotek SEC-MALS 20 and Viscotek RI Detector VE3580; Malvern Instruments) was used to determine the absolute molecular mass of proteins and protein complexes in solution. Injections of 100 μl of about 1 mg/ml material were run on a Superdex 200 10/300 GL (GE Healthcare) size-exclusion column pre-equilibrated in 50 mM HEPES (pH 8.0), 150 mM NaCl, and 1 mM TCEP at 22°C with a flow rate of 0.5 ml/min. Light scattering, refractive index (RI), and A$_{280\ nm}$ were analyzed by a homo-polymer model (OmniSEC software, v5.02; Malvern Instruments) using the following parameters for: $\partial A_{280\ nm}/\partial c = 0.97$ AU ml mg$^{-1}$ (Mis18α/β complex), 0.86 AU ml mg$^{-1}$ (His-GFP-Mis18α/His-Mis18β and His-Mis18α/His-GFP-Mis18β), 0.82 AU ml mg$^{-1}$ (Mis18α/Mis18β/Mis18BP1$_{20–130}$), 0.71 AU ml mg$^{-1}$ (Mis18α/Mis18β/His-Sumo-Mis18BP1$_{20–130}$), 1.10 AU ml mg$^{-1}$ (His-GFP-Mis18α$_{C\text{-term}}$/His-MBPMis18β$_{C\text{-term}}$), 0.86 AU ml mg$^{-1}$ (His-GFP-Mis18α$_{2M}$/His-Mis18β), $\partial n/\partial c = 0.185$ ml g$^{-1}$, and buffer RI value of 1.335. The mean standard error in the mass accuracy determined for a range of protein–protein complexes spanning the mass range of 6–600 kDa is ±1.9%.

### Construction of TetR and mCherry/mCerulean fusion constructs

Human Mis18β, Mis18BP1$_{fl}$, Mis18BP1$_{20–130}$, and Mis18BP1$_{336–483}$ were amplified from a human cDNA library (MegaMan human transcriptome library, Agilent) and cloned into the pCDNA3-mCherry or -mCerulean LIC cloning vectors (Addgene plasmids # 30125 and 30128, respectively, a gift from Scott Gradia). Mis18α was cloned into the TetR-eYFP-IRES-Puro vector. Mutant constructs, TetR-eYFP-Mis18α$_{DimerM}$, mCherry-Mis18BP1$_{20–130\ T40E}$, mCherry-Mis18BP1$_{20–130\ S110D}$, mCherry-Mis18BP1$_{20–130\ T40E/S110D}$, and mCherry-Mis18BP1$_{20–130\ T40A/S110A}$, were generated using the Quikchange mutagenesis protocol (Stratagene).

### Cell culture and transfection

The HeLa 3-8 overexpressing CENP-A-SNAP integration cell line, containing a synthetic α-satellite (alphoid) DNA array integration with tetO sites (alphoid$^{tetO}$ array) integrated in a chromosome arm, was maintained in DMEM (Gibco) supplemented with 5% fetal bovine serum (Biowest) and penicillin/streptomycin (Gibco). Cells were grown at 37°C and 5% CO$_2$. Transfections were performed in parallel with XtremeGene-9 (Roche) following manufacturer's instructions. Briefly, 24 h after plating, cells grown in 12-well plates were incubated with transfection complexes containing: 0.25 μg of each vector, 100 μl of Opti-MEM (Invitrogen), and 3 μl of Xtreme-Gene-9 reagent for 36 h.

### Immunofluorescence and quantification

Thirty-six hours after transfection, cells growing on coverslips were fixed with 2.6% paraformaldehyde in PBS 1× buffer for 10 min at 37°C. Cells were then treated with permeabilization buffer (PBS containing 0.2% Triton X-100, Sigma) for 5 min at room temperature and blocked with permeabilization buffer containing 3% BSA for 1 h at 37°C. Mouse anti-CENP-A (AN1; 1/500 dilution) and Cy-5-conjugated donkey anti-mouse secondary antibody (Jackson Immunoresearch) were used for centromere staining.

Micrographs were acquired in the Centre Optical Instrumentation Laboratory on a DeltaVision Elite system (Applied Precision) using an inverted Olympus IX-71 stand, with an Olympus

UPlanSApo 100× oil immersion objective (numerical aperture (NA) 1.4) and a Lumencor light source. Camera (Photometrics Cool Snap HQ), shutter and stage were controlled through Soft-Worx (Applied Precision). The 0.2-μm-spaced *z*-stacks were deconvolved using SoftWorx and analyzed using ImageJ software (NIH, Bethesda). For CENP-A signal quantification, a custom-made macro in ImageJ modified from Bodor *et al* [30] was used. Briefly, the CENP-A signal (Cy-5) found at the alphoid^TetO array (identified by the eYFP signal) was determined for every *z*-section within a 7-square pixel box. The mean signal intensity in the array was obtained and the background subtracted using the minimum intensity values within the section. The average intensity of the CENP-A signal from endogenous centromeres was used to normalize. At least three biological independent experiments were performed for each assay.

Expanded View for this article is available online.

## Acknowledgements

We thank the staff of the Edinburgh Protein Production Facility and the Centre Optical Instrumentation Laboratory for their help. The Wellcome Trust generously supported this work through a Wellcome Trust Career Development Grant (095822) and a Senior Research Fellowship (202811) to AAJ, a Principal Research Fellowship to WCE (073915), a Centre Core Grant (092076 and 203149) and an instrument grant (091020) to the Wellcome Trust Centre for Cell Biology, a Multi-User Equipment grant 101527/Z/13/Z to the EPPF and a Wellcome-UoE ISSF award toward the procurement of SEC-MALS equipment for the EPPF. The European Commission supported this work through a Career Integration Grant to AAJ (334291). FS is supported by the Darwin Trust of Edinburgh.

## Author contributions

AAJ conceived the project. FS, BM-P, MAA, OM, AAJ, and WCE designed experiments. FS, BM-P, MAA, MAW, and OM performed experiments and analyzed data. FS, BM-P, MAA, OM, WCE, and AAJ wrote the manuscript.

## Conflict of interest

The authors declare that they have no conflict of interest.

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
