## [Review Process File · EMBO Reports]

Manuscript EMBO-2016-43564

Molecular Basis for Cdk1-Regulated Timing of Mis18 Complex Assembly and CENP-A Deposition

Frances Spiller, Bethan Medina-Pritchard, Maria Alba Abad, Martin A. Wear, Oscar Molina,
William C. Earnshaw, and A. Arockia Jeyaprakash

Corresponding author: A. Arockia Jeyaprakash, University of Edinburgh

Review timeline:

Submission date:	24 October 2016
Editorial Decision:	24 November 2016
Revision received:	22 February 2017
Editorial Decision:	10 March 2017
Revision received:	13 March 2017
Accepted:	15 March 2017

Editor: Achim Breiling

Transaction Report:

1st Editorial Decision

24 November 2016

Thank you for the submission of your research manuscript to EMBO reports. We have now received reports from the three referees that were asked to evaluate your study, which can be found at the end of this email.

As you will see, all three referees acknowledge the potential interest of the findings. All three referees have raised a number of concerns and suggestions to improve the manuscript, or to strengthen the data and the conclusions drawn, which need to be addressed. In particular, all the concerns of the referees #1 and #2 should be addressed experimentally where possible. As the reports are below, I will not detail them here.

Given these constructive comments, we would like to invite you to revise your manuscript with the understanding that all referee concerns must be fully addressed in a complete point-by-point response. Acceptance of your manuscript will depend on a positive outcome of a second round of review. It is EMBO reports policy to allow a single round of revision only and acceptance or rejection of the manuscript will therefore depend on the completeness of your responses included in the next, final version of the manuscript.

REFeree REPORTS

Referee #1:

In this manuscript, Spiller et al use analyses both in cells and with purified proteins to describe the overall assembly of the human Mis18 complex, consisting of the paralogs Mis18alpha and Mis18beta, along with Mis18BP1. Together, these proteins are responsible for CENP-A deposition at centromeres, by recruiting HJURP/CENP-A complexes in a cell-cycle regulated manner. The authors show compelling evidence that the region of Mis18BP1 comprising residues 20-130 mediates interactions with Mis18alpha/beta, and that this interaction is mediated by phosphorylation, likely by CDKs. This is the strongest part of the paper, and represents an advance in understanding of the cell-cycle regulation of CENP-A deposition. Prior work with Mis18alpha/beta and their *S. pombe* ortholog Mis18 has shown that they possess two major domains, a central MeDiY domain and a C-terminal coiled coil, both of which can mediate protein-protein interactions. Here, the authors focus on the MeDiY domains as modulators of both oligomeric assembly and Mis18BP1 binding. While they make a number of compelling observations that run counter to established ideas of Mis18 complex architecture, the observations are not sufficiently supported or grounded in established work, and their resulting model is unsatisfying.

The use of a synthetic alphoid DNA array and tethering of Mis18alpha to this array is a powerful assay, and is used well in the paper. I would have liked to see evidence that Mis18beta was recruited to these arrays along with Mis18BP1. Secondly, the authors describe a model for cell-cycle regulation of Mis18BP1 recruitment near the end of the paper, but it seems the alphoid DNA array could have been used to test this idea: Is Mis18BP1 20-130 recruited to this array in a cell-cycle specific manner? If the authors make alanine mutations of T40 and S110 in Mis18BP1, is Mis18alpha/beta binding maintained *in vitro*? If so, is Mis18BP1 then recruited to the alphoid DNA array throughout the cell cycle?

The results in Figure 2 are unconvincing, and to my eye do not support the authors' conclusion that Mis18BP1 20-130 interacts with the Mis18alpha MeDiY domain. The size-exclusion results in Figure 2 seem to show that neither the isolated Mis18alpha or beta MeDiY domains strongly binds Mis18BP1 20-130 (the SDS-PAGE bands clearly show offset peaks in the runs where they are mixed and run together). Also, Mis18beta MeDiY seems to more significantly shift the elution volume of Mis18BP1 than does Mis18alpha (but the figure format makes this hard to discern), yet the authors discount this entirely. Based on data the authors present later (dimerization mutant of Mis18alpha), it seems more likely that Mis18BP1 interacts with Mis18alpha/beta MeDiY domain heterodimer. Did the authors try mixing the two MeDiY domains plus Mis18BP1, to see if this results in a more robust interaction?

As mentioned above, the authors show some interesting *in vitro* evidence that the Mis18alpha/beta complex forms a heterohexamer, instead of a heterotetramer as previously reported by the Foltz lab (Nardi et al). There are several problems with this section, however:

First, the authors' assertion that Nardi et al used "crude glycerol based gradient experiments" is unfair and inaccurate. That paper used a combination of size exclusion chromatography and glycerol gradient centrifugation to come up with molecular weights for Mis18alpha alone, Mis18beta alone, and the complex. While the methods may have been crude, the result was clear and made sense. Second, while the observations that Mis18alpha/beta forms a hexamer, and that two copies of Mis18BP1 bind this complex, are pretty clear, I would like to see additional supporting experiments with other tags to nail this down. For instance, putting a GFP tag on Mis18beta to see if the hexamer's MW increases by two GFP's worth. Third, as Mis18BP1 20-130 is very small, the finding that two copies of this bind would be strengthened by using either a longer Mis18BP construct or one with a GFP tag.

There are several reasons for skepticism regarding the models presented in Figure 5 for the assembly of the full heterohexameric complex:

- First, in both models, Mis18alpha and Mis18beta MeDiY domains must preferentially form heterodimers rather than alpha-alpha or beta-beta homodimers. From the gel filtration results in Figure 2, it rather looks like beta might form a nice dimer in isolation. But, MeDiY dimerization propensities were not examined. As the authors have the proteins, they should do SEC-MALS analysis on them to establish homo- and heterodimerization propensities.
- Second, both models also depend on the C-terminal alpha-helical domains making interactions that run counter to the findings of Nardi et al., who suggested that they form an alpha₂-beta₂ tetramer.

Nardi's results were not complete/convincing, though, so the authors have a chance to correct the record regarding the role of the C-terminal domains. To claim that they have revealed "the molecular basis for the assembly and regulation of the Mis18 complex", this should be addressed. - Third, in both models presented in Figure 5, there is no reason to think that, in the absence of Mis18bp1 binding, that these would not simply form continuous oligomeric complexes of indeterminate size, rather than a clean hexamer. That is, what's stopping the "free" mis18alpha subunits from dimerizing with more mis18beta?

I think the results with the Mis18alpha 2M mutants might be better explained in the context of Mis18BP1 binding to a Mis18alpha/beta MeDiY domain heterodimer. Can the authors address this? They seem to have missed several opportunities to test just such an idea, for example by mixing the proteins used for Figure 2 together to see if Mis18BP1 binding is more robust.

Minor points:

Despite the evident degradation of mMis18alpha MeDiY in Figure EV2, this result seems cleaner than when using the GFP fusions. Could these GFP fusions also be interfering with Mis18BP1 binding?

Figure 2B is missing the key for the line colors in the upper panel.

In figure 3, the wavy pattern of molecular weight estimate across the peak is a classic characteristic of either a baseline issue or, more likely, a problem with calibration of the "band broadening" parameters. I would try to adjust these to get a flatter molecular weight estimate across the peak. This will not change the final molecular weight numbers by more than a couple percent, so the conclusions from these experiment still stand.

The use of "2M" and "4M" for the mutants is a bit confusing - I'd recommend using more descriptive names such as "delta-dim" for the 2M (removal of dimerization) and "delta-sub" for the 4M (removal of substrate binding)

The result in Figure 4B, while it may be accurate, is quite weak. Also, is it true that the 2M mutant is expressed at much lower levels than the other two constructs?

Referee #2:

In this manuscript, the authors investigate the biochemical basis for centromere propagation. Deposition of centromere-defining CENP-A nucleosome is tightly controlled during the cell cycle, but the organization and associations of the Mis18 complex that directs this assembly process were incompletely understood. Using a nice combination of cell biological and biochemical approaches, the authors present a compelling biochemical rationale for how the organization of this complex and how specific CDK phosphorylation events in a conserved region of MIS18bp1 destabilize the Mis18 holo-complex. I fully support publication of this paper, but I do have some suggestions for improving the presentation and providing some additional controls as detailed below.

1. Figure 4B: The authors perform a co-IP experiment to test whether mutating residues in the putative dimerization interface affects the ability of MIS18alpha to bind to MIS18BP1. Based on the data that is presented, I don't find this gel completely convincing as the difference is subtle. It also appears that there is less protein in the input for Mis18alpha2M. This raises that possibilities that the mutations introduced into Mis18a2M disrupt protein structure and folding in a broader manner and make this experiment harder to interpret. Many of the other binding assays were conducted using gel filtration, and it is not clear why the co-IP was done instead here.

2. For Figure 1 and 3, the potential for self-oligomerization of Mis18alpha and beta is not tested. The authors arrive at a model of a core heterotrimer (2 alpha, 1 beta), which incorporates into a heterohexameric holocomplex. To fully understand this behavior, it would be helpful to see the SEC and MALS data for Mis18 alpha and beta by themselves. The authors propose that only Mis18alpha, but not beta can bind Mis18bp1, and that the number of Mis18alpha molecules in the complex limits

the number of associated Mis18bp1 molecules to 2. They could test this directly by assembling the complex with only Mis18alpha, but not beta, and then testing its stoichiometry. Does it bind more Mis18bp1? Or does it fail to assemble without both Mis18 paralogs?

3. Figure 2C: without seeing the migration of MIS18betaMeDIY by itself, it is hard to determine whether or not it interacts with Mis18BP120-130. The authors argue that it doesn't, but the data as they stand now don't demonstrate this convincingly.

4. The authors do a good job of citing the prior literature on the Mis18 complex. However, they do not adequately mention the related work by McKinley (PMID: 25036634) that demonstrated that the N-terminus of MIS18BP1 associates with Mis18 alpha/beta and that this interaction is negatively regulated by CDK phosphorylation. The existence of this prior work does not diminish the importance of the solid biochemistry in this paper, but should be mentioned directly.

Referee #3:

In the manuscript titled "Molecular basis for cell cycle control of Mis18 Complex assembly...", Spiller et al. use *in vitro* biochemistry to demonstrate that 4 Mis18a's, 2 Mis18b's, and 2 Mis18BP1's assemble in a phospho-regulated manner. They argue that this is the functional complex, and thus is mechanistic foundation for centromere assembly. This work is well executed and the problem of CENP-A loading is both critical and unresolved. Previous works have shown that Mis18 complex localization to centromeres is the first step in loading new CENP-A, as it recruits the HJURP chaperone which assembles CENP-a nucleosomes. Localization of the Mis18 complex to centromeres is negatively regulated by CDK phosphorylation (Jansen lab), as inhibition of CDK1 and 2 in G2 causes inappropriate recruitment and CENP-A loading. Interestingly, the Mis18 Complex is also thought to be positively regulated by Plk phosphorylation in anaphase (Cheeseman Lab). Here the authors find that CDK negatively regulates complex formation by inhibiting Mis18BP1 binding to the mis18 subunits. The work is largely *in vitro* using purified proteins and is supplemented by an artificial cell based assay, the lac array recruitment assay.

While overall this is an interesting study that, as I mentioned, is well executed, my enthusiasm is damped by two major factors.

1) The work does not probe endogenous proteins in any fashion including by knockdown/replace. In my opinion, overexpressing proteins in the cell for colocalization to an artificial array is virtually the same as *in vitro* pull down experiments and does not add much to the paper. I also find it troubling that the overexpressed proteins (even the controls) fail to localize to endogenous centromeres. This raises strong doubt in my mind about the validity of the work.

2) If I assume the results are true from the experiments, which I question, I do not see precisely how this work pushes our field forward. This is somewhat incremental, and while not unimportant, does not really change how I previously thought about this problem. I totally agree that it adds some important details, but would want more mechanistic information such as how mutations affect rate constants of CENP-A assembly or are the mutations dominant?

In my opinion, these two major issues preclude recommendation for publication at this time.

1st Revision - authors' response

22 February 2017

Thank you very much for giving us the opportunity to answer the reviewers' comments. We are pleased that the reviewers agree on finding our study important and compelling. We have considered the reviewers' comments carefully and revised our manuscript accordingly by performing a number of additional experiments. Particularly, we have strengthened our conclusion that the Mis18 complex is a hetero-octamer made of 4 Mis18, 2 Mis18 and 2 Mis18BP1 by performing additional SEC-MALS experiments (by introducing/swapping bulkier tags on different protein subunits). By carrying out a series of Size Exclusion Chromatography (SEC) and Ni-NTA

pull-down experiments, we show that while Mis18 MeDiY can bind Mis18BP1 on its own, Mis18 MeDiY/ MeDiY heterodimer binds Mis18BP1 more robustly. Based on this observation, we propose a model where the Mis18 / hetero-hexamers (4 Mis18 : 2 MeDiY) binds 2 copies of Mis18BP1 via its Mis18 / MeDiY hetero-dimers and the remaining two copies of Mis18 MeDiY form a homo-dimer. We have also analysed the recruitment of mCherry-Mis18BP1 to the alphoidtetO array by TetR-eYFP-Mis18 during early stages of mitosis (prophase-anaphase onset) and show that Mis18BP1 is recruited to the array in a cell cycle specific manner. Moreover, the Mis18BP1 mutant that is constitutively present in the Mis18 complex does not show such cell cycle specific targeting and remains stably associated with centromeres. This unambiguously confirms that the timing of Mis18 complex assembly is determined by the Cdk1 phosphorylation of Mis18BP1 residues that we identified to be critical for Mis18 / binding. Overall, we feel that the critique and suggestions provided by the reviewers improved our manuscript and we hope that you find our revised manuscript suitable for publication.

 The point-by-point response to the reviewer's comments is as follows:

Referee #1:

In this manuscript, Spiller et al use analyses both in cells and with purified proteins to describe the overall assembly of the human Mis18 complex, consisting of the paralogs Mis18alpha and Mis18beta, along with Mis18BP1. Together, these proteins are responsible for CENP-A deposition at centromeres, by recruiting HJURP/CENP-A complexes in a cell-cycle regulated manner. The authors show compelling evidence that the region of Mis18BP1 comprising residues 20-130 mediates interactions with Mis18alpha/beta, and that this interaction is mediated by phosphorylation, likely by CDKs. This is the strongest part of the paper, and represents an advance in understanding of the cell-cycle regulation of CENP-A deposition. Prior work with Mis18alpha/beta and their *S. pombe* ortholog Mis18 has shown that they possess two major domains, a central MeDiY domain and a C-terminal coiled coil, both of which can mediate protein-protein interactions. Here, the authors focus on the MeDiY domains as modulators of both oligomeric assembly and Mis18BP1 binding. While they make a number of compelling observations that run counter to established ideas of Mis18 complex architecture, the observations are not sufficiently supported or grounded in established work, and their resulting model is unsatisfying.

The use of a synthetic alphoid DNA array and tethering of Mis18alpha to this array is a powerful assay, and is used well in the paper. I would have liked to see evidence that Mis18beta was recruited to these arrays along with Mis18BP1.

In agreement with the reviewer, we have expressed TetR-eYFP-Mis18α in HeLa 3-8 cells containing the alphoid^{tetO} array integrated in a chromosome arm and analyzed the ability of Mis18α to recruit mCherry-Mis18BP1₂₀₋₁₃₀ and mCerulean-Mis18β to the same tethering site. The results show that Mis18α recruits both Mis18BP1 and Mis18β to the array. We have now added the quantification and the corresponding representative images to Fig EV1B and changes to the text on page 5.

Secondly, the authors describe a model for cell-cycle regulation of Mis18BP1 recruitment near the end of the paper, but it seems the alphoid DNA array could have been used to test this idea: Is Mis18BP1 20-130 recruited to this array in a cell-cycle specific manner? If the authors make alanine mutations of T40 and S110 in Mis18BP1, is Mis18alpha/beta binding maintained in vitro? If so, is Mis18BP1 then recruited to the alphoid DNA array throughout the cell cycle?

We thank the reviewer for raising this point. We have now quantified the signal of mCherry-Mis18BP1₂₀₋₁₃₀ at the array in cells expressing TetR-eYFP-Mis18α_{wt} during different stages of mitosis. The results show that recruitment of Mis18BP1₂₀₋₁₃₀ to the array is impaired during early stages of mitosis (prophase-prometaphase-metaphase), whilst a non-phosphorylatable mutant Mis18BP1₂₀₋₁₃₀ T40A/S110A constitutively associated with the array throughout the cell cycle. This data indicates that Mis18α tethered to the array recruits Mis18BP1 in a cell cycle dependent manner. We have added the quantification and corresponding representative images to Fig 5D and changes to the text on page 13. We also show that Mis18BP1₂₀₋₁₃₀ T40A/S110A can bind Mis18α/β in vitro as analyzed by SEC (data included in Fig EV5).

The results in Figure 2 are unconvincing, and to my eye do not support the authors' conclusion that Mis18BP1 20-130 interacts with the Mis18alpha MeDiY domain. The size-exclusion results in Figure 2 seem to show that neither the isolated Mis18alpha or beta MeDiY domains strongly binds Mis18BP1 20-130 (the SDS-PAGE bands clearly show offset peaks in the runs where they are mixed and run together). Also, Mis18beta MeDiY seems to more significantly shift the elution volume of Mis18BP1 than does Mis18alpha (but the figure format makes this hard to discern), yet the authors discount this entirely. Based on data the authors present later (dimerization mutant of Mis18alpha), it seems more likely that Mis18BP1 interacts with Mis18alpha/beta MeDiY domain heterodimer. Did the authors try mixing the two MeDiY domains plus Mis18BP1, to see if this results in a more robust interaction?

We thank this reviewer for raising these concerns. The SEC experiments originally shown in Fig 2 were performed using two different SEC columns (His-GFP-Mis18 α _{MeDiY}/Mis18BP1₂₀₋₁₃₀ using Superdex 200 increase 10/300 and His-GFP-Mis18 β _{MeDiY}/Mis18BP1₂₀₋₁₃ using Superdex 75 10/300) and hence the elution profiles from these experiments cannot be directly compared. To clarify this, we have now repeated SEC analyses of Mis18BP1₂₀₋₁₃₀ mixed with untagged Mis18 α _{MeDiY} and Mis18 β _{MeDiY} in separate experiments using a Superdex 75 10/300 column. Mis18 α _{MeDiY} eluted earlier in the presence of Mis18BP1₂₀₋₁₃ as compared to its elution volume on its own confirming interaction (revised Fig EV2A). On the contrary, Mis18 β _{MeDiY} and Mis18BP1₂₀₋₁₃ eluted at distinct elution volumes showing no detectable interaction (new Fig 2C). We also noticed that by increasing protease inhibitors and by extensive cleaning of the size exclusion column to get rid of any trace amount of proteases we could avoid Mis18 α _{MeDiY} degradation (seen originally in Fig EV2). As MWs of Mis18 α _{MeDiY} (12.4 kDa) and Mis18BP1₂₀₋₁₃₀ (12.7 kDa) are almost identical these migrated similarly in the SDS-PAGE (Fig EV2A). To further strengthen Mis18 α _{MeDiY}-Mis18BP1₂₀₋₁₃₀ interaction, we repeated SEC analysis of His-GFP-Mis18 α _{MeDiY} mixed with Mis18BP1₂₀₋₁₃₀ (using Superdex 200 increase 10/300). Whilst His-GFP-Mis18 α _{MeDiY} on its own eluted at 13.8 ml, the His-GFP-Mis18 α _{MeDiY}/Mis18BP1₂₀₋₁₃₀ complex eluted at 13.3 ml (Fig 2B). By decreasing sample injection volumes and additional cleaning of the SEC column we could improve the SEC resolution as compared to the data shown in original Fig 2B. Overall, this confirms that the Mis18 α _{MeDiY} domain can directly interact with Mis18BP1₂₀₋₁₃₀.

We also agree that testing the ability of Mis18 α _{MeDiY}/Mis18 β _{MeDiY} hetero-dimer to bind Mis18BP1₂₀₋₁₃₀ is important to fully understand the molecular basis of Mis18 complex assembly. We have now addressed these concerns by performing Ni-NTA pull-down assays at varying salt concentrations to assess the relative Mis18BP1 binding strengths of Mis18 α _{MeDiY} and Mis18 α _{MeDiY}/Mis18 β _{MeDiY}. Our results show that while Mis18 α _{MeDiY} can directly interact with Mis18BP1₂₀₋₁₃₀, the Mis18 α _{MeDiY}/Mis18 β _{MeDiY} hetero-dimer binds Mis18BP1₂₀₋₁₃₀ more robustly (as shown in Fig 2D). We have now explained this on page 7 under the heading "Mis18BP1₂₀₋₁₃₀ binds Mis18 α _{MeDiY}/ β _{MeDiY} hetero-dimer more robustly than Mis18 α _{MeDiY}".

As mentioned above, the authors show some interesting in vitro evidence that the Mis18alpha/beta complex forms a heterohexamer, instead of a heterotetramer as previously reported by the Foltz lab (Nardi et al). There are several problems with this section, however:

First, the authors' assertion that Nardi et al used "crude glycerol based gradient experiments" is unfair and inaccurate. That paper used a combination of size exclusion chromatography and glycerol gradient centrifugation to come up with molecular weights for Mis18alpha alone, Mis18beta alone, and the complex. While the methods may have been crude, the result was clear and made sense.

We would like to clarify that the word "crude" was used in reference to the technique, not the manner in which these experiments were performed. However, in agreement with the reviewer we have removed the word "crude" and the sentence now reads, "glycerol based gradient experiments."

Second, while the observations that Mis18alpha/beta forms a hexamer, and that two copies of Mis18BP1 bind this complex, are pretty clear, I would like to see additional supporting experiments with other tags to nail this down. For instance, putting a GFP tag on Mis18beta to see if the hexamer's MW increases by two GFP's worth.

We agree with the reviewer that using different tags on different proteins would further strengthen our conclusions on the subunit stoichiometry of the Mis18 complex. We have now performed SEC-MALS with His-Mis18 α /His-GFP-Mis18 β and the result is consistent with our original data that Mis18 α/β is a hetero-hexamers with a 4:2 stoichiometry (revised Fig 3C and EV3C). We have amended the text on page 9 to include this data.

Third, as Mis18BP1 20-130 is very small, the finding that two copies of this bind would be strengthened by using either a longer Mis18BP construct or one with a GFP tag.

We have also carried out SEC-MALS analysis of the Mis18 complex reconstituted with His-SUMO-Mis18BP1₂₀₋₁₃₀ and show unambiguously that the Mis18 complex is a hetero-octamer with 2 copies of Mis18BP1 (revised Fig 3G, EV3F and page 10).

There are several reasons for skepticism regarding the models presented in Figure 5 for the assembly of the full heterohexameric complex:

First, in both models, Mis18 α and Mis18 β MeDiY domains must preferentially form heterodimers rather than alpha-alpha or beta-beta homodimers. From the gel filtration results in Figure 2, it rather looks like beta might form a nice dimer in isolation. But, MeDiY dimerization propensities were not examined. As the authors have the proteins, they should do SEC-MALS analysis on them to establish homo- and heterodimerization propensities.

The propensities of the MeDiY domains to dimerize was addressed in our previous publication, Subramanian et al., 2016 - PMID: [PMC4818781](https://pubmed.ncbi.nlm.nih.gov/2718781/): Mis18 α_{MeDiY} , on its own forms a homo-dimer but prefers to form a hetero-dimer in the presence of Mis18 β_{MeDiY} , whilst Mis18 β_{MeDiY} is a monomer on its own. We have now cited our previous work to clarify this on page 7.

Second, both models also depend on the C-terminal alpha-helical domains making interactions that run counter to the findings of Nardi et al., who suggested that they form an alpha₂-beta₂ tetramer. Nardi's results were not complete/convincing, though, so the authors have a chance to correct the record regarding the role of the C-terminal domains. To claim that they have revealed "the molecular basis for the assembly and regulation of the Mis18 complex", this should be addressed.

In agreement with this reviewer's suggestion, we reconstituted the Mis18 α/β C-terminal helical assembly using individually purified His-GFP-Mis18 α_{C-term} (Mis18 α 188-end) and His-MBP-Mis18 β_{C-term} (Mis18 β 184-end) and analyzed their composition using SEC-MALS. The measured MW of His-GFP-Mis18 α_{C-term} /His-MBP-Mis18 β_{C-term} complex (115.5 ± 2.2 kDa) matches with a calculated MW of a hetero-trimeric assembly composed of 2 His-GFP-Mis18 α_{C-term} and 1 His-MBP-Mis18 β_{C-term} (119.6 kDa) (Fig 3D and EV3D). This implies that the formation of the full length hetero-hexameric Mis18 α/β assembly requires further oligomerization of Mis18 α/β hetero-trimers mediated by Mis18 $\alpha_{MeDiY}/\beta_{MeDiY}$ hetero-dimers and strengthens the model shown in Fig 3E. To accommodate these new findings we have now added a section "Mis18 α/β hetero-hexamer is assembled from hetero-trimers of C-terminal α -helical domains and hetero-dimers of MeDiY domains" (page 9).

Third, in both models presented in Figure 5, there is no reason to think that, in the absence of Mis18bp1 binding, that these would not simply form continuous oligomeric complexes of indeterminate size, rather than a clean hexamer. That is, what's stopping the "free" mis18 α subunits from dimerizing with more mis18 β ?

In our revised model (Fig 4A), 2 copies of Mis18BP1 bind to 2 copies of Mis18 $\alpha_{MeDiY}/\beta_{MeDiY}$ hetero-dimer and we hypothesize that the remaining 2 copies of Mis18 α_{MeDiY} form a homo-dimer based on its ability to self-oligomerize (as we had shown before in Subramanian et al., EMBO rep 2016).

I think the results with the Mis18alpha 2M mutants might be better explained in the context of Mis18BP1 binding to a Mis18alpha/beta MeDiY domain heterodimer. Can the authors address this? They seem to have missed several opportunities to test just such an idea, for example by mixing the proteins used for Figure 2 together to see if Mis18BP1 binding is more robust.

We thank the reviewer for this suggestion. As detailed in our response to one of the earlier queries (query no. 3), we have now tested if Mis18 α _{MeDiY}/Mis18 β _{MeDiY} hetero-dimer can bind Mis18BP1₂₀₋₁₃₀ using SEC (Fig EV2B) and Ni-NTA pull-down experiments (Fig 2D). We show that Mis18 α _{MeDiY}/Mis18 β _{MeDiY} hetero-dimer binds Mis18BP1₂₀₋₁₃₀ more robustly as compared to Mis18 α _{MeDiY} on its own. In the light of this data, we think the MeDiY dimer-disrupting mutation not only affects the overall oligomeric structure of the Mis18 α / β oligomer, but also Mis18BP1 binding. We have addressed this appropriately in the revised manuscript (on page 7-8 and 10-11).

Minor points:

Despite the evident degradation of mMis18alpha MeDiY in Figure EV2, this result seems cleaner than when using the GFP fusions. Could these GFP fusions also be interfering with Mis18BP1 binding?

By decreasing the sample injection volume and extensive cleaning of size exclusion column we could improve SEC resolution of His-GFP-Mis18 α _{MeDiY}/Mis18BP1₂₀₋₁₃₀: His-GFP- Mis18 α _{MeDiY} eluted at 13.3 ml in the presence of Mis18BP1₂₀₋₁₃₀ as compared to 13.8 ml when on its own. This together with the new Ni-NTA pull-down data shown in revised Fig 2D, unambiguously show that His-GFP-Mis18 α _{MeDiY} can directly interact with Mis18BP1₂₀₋₁₃₀ and GFP tag does not interfere with Mis18BP1 binding.

Figure 2B is missing the key for the line colors in the upper panel.

We thank the reviewer for bring this to our attention. We have now color-coded labels matching their respective chromatogram.

In figure 3, the wavy pattern of molecular weight estimate across the peak is a classic characteristic of either a baseline issue or, more likely, a problem with calibration of the "band broadening" parameters. I would try to adjust these to get a flatter molecular weight estimate across the peak. This will not change the final molecular weight numbers by more than a couple percent, so the conclusions from these experiment still stand.

As this reviewer has correctly stated, the pattern and behavior (wavy mass distribution) seen in Fig 3 is often the result of issues with the baseline and/or the detector calibration parameters - Sigma (band broadening) and Tau (tailing). These have been adjusted and optimized as much as possible (Sigma values are 0.2 or below and corresponding Tau values are smaller) and the profiles replotted to give cleaner mass distributions across the peaks in Figure 3.

The use of "2M" and "4M" for the mutants is a bit confusing - I'd recommend using more descriptive names such as "delta-dim" for the 2M (removal of dimerization) and "delta-sub" for the 4M (removal of substrate binding)

In order to clarify the nomenclature of the mutants, we have now changed the naming of "2M" and "4M" to "DimerM" and "PocketM" respectively, in the text and figures.

The result in Figure 4B, while it may be accurate, is quite weak. Also, is it true that the 2M mutant is expressed at much lower levels than the other two constructs?

In order to strengthen this conclusion we have now replaced this data with a new Ni-NTA pull-down assay where purified recombinant His-GFP-Mis18 α /His-Mis18 β complexes harboring dimer-disrupting and pocket mutations were tested for their ability to bind Mis18BP1₂₀₋₁₃₀. The data shows that the Mis18 α / β complex harboring the dimer-disrupting mutation failed to interact with Mis18BP1₂₀₋₁₃₀ (Fig 4C).

Referee #2:

In this manuscript, the authors investigate the biochemical basis for centromere propagation. Deposition of centromere-defining CENP-A nucleosome is tightly controlled during the cell cycle, but the organization and associations of the Mis18 complex that directs this assembly process were incompletely understood. Using a nice combination of cell biological and biochemical approaches, the authors present a compelling biochemical rationale for how the organization of this complex and how specific CDK phosphorylation events in a conserved region of MIS18bp1 destabilize the Mis18 holo-complex. I fully support publication of this paper, but I do have some suggestions for improving the presentation and providing some additional controls as detailed below.

1. Figure 4B: The authors perform a co-IP experiment to test whether mutating residues in the putative dimerization interface affects the ability of MIS18alpha to bind to MIS18BP1. Based on the data that is presented, I don't find this gel completely convincing as the difference is subtle. It also appears that there is less protein in the input for Mis18alpha2M. This raises the possibilities that the mutations introduced into Mis18a2M disrupt protein structure and folding in a broader manner and make this experiment harder to interpret. Many of the other binding assays were conducted using gel filtration, and it is not clear why the co-IP was done instead here.

We unfortunately could not use SEC to address this as Mis18a on its own does not behave as a monodisperse sample. Moreover, we believe that evaluating the role MeDiY dimerization interface and putative substrate-binding pocket of Mis18a in Mis18BP1₂₀₋₁₃₀ binding needs to be carried out in the context of the Mis18a/β complex rather than in isolation. Hence, we have now performed a new Ni-NTA pull-down assay using purified recombinant His-GFP-Mis18a/His-Mis18β complexes with or without dimer-disrupting and pocket mutations. Use of purified recombinant protein in this pull-down assay also allowed us to better control the amount of proteins used, a concern raised by this reviewer. This experiment now clearly shows that the Mis18a/β complex containing dimer-disrupting mutation cannot bind to Mis18BP1₂₀₋₁₃₀ (Fig 4C).

2. For Figure 1 and 3, the potential for self-oligomerization of Mis18alpha and beta is not tested. The authors arrive at a model of a core heterotrimer (2 alpha, 1 beta), which incorporates into a heterohexameric holocomplex. To fully understand this behavior, it would be helpful to see the SEC and MALS data for Mis18 alpha and beta by themselves. The authors propose that only Mis18alpha, but not beta can bind Mis18bp1, and that the number of Mis18alpha molecules in the complex limits the number of associated Mis18bp1 molecules to 2. They could test this directly by assembling the complex with only Mis18alpha, but not beta, and then testing its stoichiometry. Does it bind more Mis18bp1? Or does it fail to assemble without both Mis18 paralogs?

We would like to thank the reviewer for these suggestions. Unfortunately, we could not perform SEC-MALS analysis of Mis18a either on its own or in complex with Mis18BP1₂₀₋₁₃₀, mainly because Mis18a does not behave as a monodisperse sample when not in complex with Mis18β. However, to strengthen the conclusions that Mis18a/β core complex is a hetero-trimer, we have now carried out additional SEC-MALS analysis of the Mis18a/β C-terminal helical assembly (His-GFP-Mis18a_{C-term}/His-MBP-Mis18β_{C-term}). The result shows that the C-terminal α-helical domains of Mis18a and Mis18β form a hetero-trimer consisting of 2 Mis18a and 1 Mis18β and requires hetero-dimerization of Mis18a_{MeDiY} and Mis18β_{MeDiY} for the assembly of the full length hetero-hexameric Mis18a/β complex (Fig 3D and EV3D).

3. Figure 2C: without seeing the migration of MIS18betaMeDIY by itself, it is hard to determine whether or not it interacts with Mis18BP120-130. The authors argue that it doesn't, but the data as they stand now don't demonstrate this convincingly.

We have now analyzed the ability of Mis18β_{MeDiY} to interact with Mis18BP1₂₀₋₁₃₀ using purified untagged proteins in SEC (new Fig 2C). In addition, we have also performed Ni-NTA pull-down assay to test the same (new Fig 2D). The outcome of these experiments shows that Mis18β_{MeDiY} does not interact with Mis18BP1₂₀₋₁₃₀.

4. The authors do a good job of citing the prior literature on the Mis18 complex. However, they do not adequately mention the related work by McKinley (PMID: 25036634) that demonstrated that the N-terminus of MIS18BP1 associates with Mis18 alpha/beta and that this interaction is negatively regulated by CDK phosphorylation. The existence of this prior work does not diminish the importance of the solid biochemistry in this paper, but should be mentioned directly.

Thank you for raising this point. We have now additionally cited this key relevant paper in support of the involvement of Mis18BP1 N-terminal region for Mis18 complex formation on page 5 and the Cdk1 mediated regulation of Mis18 complex assembly on page 4,12 and 13.

Referee #3:

In the manuscript titled "Molecular basis for cell cycle control of Mis18 Complex assembly...", Spiller et al. use in vitro biochemistry to demonstrate that 4 Mis18a's, 2 Mis18b's, and 2 Mis18BP1's assemble in a phospho-regulated manner. They argue that this is the functional complex, and thus is mechanistic foundation for centromere assembly. This work is well executed and the problem of CENP-A loading is both critical and unresolved. Previous works have shown that Mis18 complex localization to centromeres is the first step in loading new CENP-A, as it recruits the HJURP chaperone which assembles CENP-a nucleosomes. Localization of the Mis18 complex to centromeres is negatively regulated by CDK phosphorylation (Jansen lab), as inhibition of CDK1 and 2 in G2 causes inappropriate recruitment and CENP-A loading. Interestingly, the Mis18 Complex is also thought to be positively regulated by Plk phosphorylation in anaphase (Cheeseman Lab). Here the authors find that CDK negatively regulates complex formation by inhibiting Mis18BP1 binding to the mis18 subunits. The work is largely in vitro using purified proteins and is supplemented by an artificial cell based assay, the lac array recruitment assay.

While overall this is an interesting study that, as I mentioned, is well executed, my enthusiasm is damped by two major factors:

1) The work does not probe endogenous proteins in any fashion including by knockdown/replace. In my opinion, overexpressing proteins in the cell for colocalization to an artificial array is virtually the same as in vitro pull down experiments and does not add much to the paper. I also find it troubling that the overexpressed proteins (even the controls) fail to localize to endogenous centromeres. This raises strong doubt in my mind about the validity of the work.

We are glad that this reviewer finds our study interesting and well executed. We would like to highlight that the TetO/LacO array-based tethering assays are a well established and powerful tool in the centromere field (as acknowledged by Reviewer 1) to dissect roles of specific proteins and/or protein interactions on CENP-A deposition at a molecular level in cells (Nakano et al., Dev Cell, 2008; Mendiburo et al., Science, 2011; Ohzeki et al., EMBO J, 2012; Zasadzinska et al, EMBO J, 2013; Hori et al., Dev Cell., 2014; Chen et al., JCB, 2014; Shono et al., JCS, 2015; Tachiwana et al., Cell Rep, 2015; Logsdon et al., JCB, 2015; Nardi et al., Mol Cell, 2016; Ohzeki et al., Dev Cell, 2016; Martins et al., Mol Biol Cell, 2016; Stellfox et al., Cell Rep, 2016; Molina et al., Nat Commun, 2016).

We have used this assay not only to validate protein interactions but also to evaluate their function by analyzing their ability to deposit CENP-A at the tethering site. Moreover, our new data included in revised Fig 5D convincingly show the cell cycle-dependent regulation of Mis18BP1 recruitment to the array. We do not generally see the overexpressed proteins at endogenous centromeres, possibly for the following reasons: i) Mis18 proteins localize at centromeres during a narrow time window (late telophase – early G1) and ii) TetR-eYFP-Mis18a is likely to prefer and concentrate at the aliphoid^{tetO} array over endogenous centromeres due its stronger affinity for the array. However, we do see the over expressed proteins localizing at endogenous centromeres in a small percentage of cells showing that the TetR-eYFP-Mis18a is capable of associating with endogenous centromeres (data not shown).

2) If I assume the results are true from the experiments, which I question, I do not see precisely how this work pushes our field forward. This is somewhat incremental, and while not unimportant, does not really change how I previously thought about this problem. I totally agree that it adds some important details, but would want more mechanistic information such as how mutations affect rate

constants of CENP-A assembly or are the mutations dominant? In my opinion, these two major issues preclude recommendation for publication at this time.

As Reviewer 1 and 2 had noted, we had presented compelling evidences for the conclusions drawn in the original manuscript. We have now further strengthened our conclusions by doing a series of additional experiments and feel that the revised manuscript is much more convincing and solid. Increasing evidences emphasize 'protein-oligomerization' as an emerging regulatory principle to control the inherently complex process of centromere establishment and maintenance: HJURP dimerization is required for stable CENP-A deposition at centromeres (Zasadzińska et al., 2013); CENP-C possesses a well conserved cupin domain (Mif2 homology domain) that dimerizes and is required for centromere association (Caroll et al., 2010; Trazzi et al., 2009; Cohen et al., 2008); Oligomerization of Mis18 proteins are essential for centromere association and function (Subramanian et al., 2016; Nardi et al., 2016). Hence we believe that it is of paramount importance to understand the precise molecular architecture of these protein assemblies and their functional relevance in order to obtain the mechanistic understanding of centromere establishment and maintenance. Along this line, we, as Reviewer 1 and 2 had rightly recognized, strongly feel that our work presented here providing molecular insights into the architecture and regulation of the Mis18 complex makes crucial contribution to our understanding of this essential biological process.

2nd Editorial Decision

10 March 2017

Thank you for the submission of your revised manuscript to our editorial offices. We have now received the reports from the referees that were asked to re-evaluate your study (you will find enclosed below). As you will see, referees #1 and #2 now fully support the publication of your manuscript in EMBO reports (after minor revision of the manuscript text). Referee #3 is still very critical and does not support publication. Nevertheless, as two referees support publication, and also in the light of a recent paper that is fully in agreement of your findings, we decided to proceed with your manuscript. However, we ask you to address the suggestions of referees #1 and #2 in a final revised version. In particular, please cite the related study properly and also add a paragraph to the discussion commenting on the other work and highlighting similarities and/or differences to your results (as indicated by referee #2).

Further, I also have a few editorial requests:

The title of the paper is presently a bit convoluted. Could you provide a simpler title?

EV Figs. 2-5 are in landscape format. Please submit these as portrait. Please refer to: http://embopress.org/sites/default/files/EMBOPress_Figure_Guidelines_061115.pdf

Please add scale bars to all microscopic images (i.e. the panels in Figs. 1, 4, 5 & EV1).

There is clearly a splice in the Western in Fig. EV2B. Please add a vertical dividing line, indicating that these are two different blots. If in other panels also images of different blots were put together, please also add dividing lines.

It seems that some images in Figs. 1B, 4E & 5C appear more than once. Please indicate this in the legend (that they are from the same experiment).

Were any of these data submitted to a database? If yes, please indicate this in the methods section.

REFEREE REPORTS

Referee #1:

The revised manuscript does an excellent job addressing all of my criticisms of the earlier draft. Their revised model, which is extremely well-supported by their new data, also agrees well with a model reported in eLife by the Musacchio lab after this paper was initially submitted. I think this

manuscript should be published essentially as-is. I have a couple notes the authors may wish to address, but I certainly don't need to see the manuscript again.

Notes:

In the section titled "The Mis18alpha MeDiY domain directly interacts with MIS18BP1 20-130 in vitro", the authors should mention the the different behavior of the isolated MeDiY domains of Mis18alpha and Mis18beta is because of alpha's tendency to form homodimers, in contrast to the lack of homodimerization in Mis18beta.

Figure 3A does not have a calculated molecular weight noted in parentheses. This may or may not have been intentional.

Referee #2:

Through this revised manuscript, the authors have successfully addressed my key comments and concerns from the previous version of the paper. Based on their changes, and on their explanations to some of my comments, I now find this paper suitable for publication. My one final lingering concern is related to the recent paper from the Musacchio lab (reference 30). There is significant overlap between these papers. I don't think that this is a problem, as two distinct labs establishing this important scientific point definitely helps this field. In addition, this current paper was clearly significantly through the publication process before the publication of the Musacchio lab work. However, I was disappointed by the way that the Musacchio paper was cited, in which the authors essentially added this as a note added in proof, even though they are still able to make substantial changes to their paper. I would much prefer this to be cited more clearly in the discussion, including directly commenting on similarities and potentially any subtle differences between the papers. It may also be good to cite the Musacchio paper at the end of the introduction, or at selected points within the results.

Referee #3:

In the revised manuscript, the authors have, in parts, attempted to address concerns raised by this reviewer by modifying the text and adding new experimentation (as requested by the other two reviewers). Unfortunately, the authors did not address my two concerns and thus I cannot recommend publication at this time. Please see below for detailed comments:

1) My first point was that the tethering system alone cannot reveal cellular function. The authors rebut by citing many paper which have employed this technology and note in data not shown that the proteins of interest localize to endogenous centromeres. I will note that in the cited papers, the tethering system is meant as a bolstering approach and nearly all of those works probe endogenous proteins at endogenous centromeres at some point in the paper. Thus I feel that these papers in fact bolster my criticism. In the end, my comment remains unchanged, please refer to original review.

2) The authors argue that their work is indeed impactful and mechanistic. I apologize for implying that it was not mechanistic, it is. However I feel that the work adds precision to an already appreciated cellular mechanism. This is subject to opinion and I will leave the editor to decide that. However I also asked that the authors make mutations to endogenous proteins to show that the rates, timing, or levels of CENP-A loading were impacted by these mutations at centromeres. The authors seemingly ignored that comment and therefore I have no recourse but to request it, or at least a reasonable argument, again.

2nd Revision - authors' response

13 March 2017

Thank you very much for getting our revised manuscript re-reviewed. We are glad that Reviewer 1 and 2 fully support the publication of this manuscript in EMBO reports.

We have now made additional changes as requested by you and Reviewer 1 and 2. As you will see, the revised manuscript discusses the recently published work from the Musacchio lab in the last paragraph (Pages 12 and 13 of the revised manuscript).

We have addressed the editorial requests as listed below:

1. Modified the title: the revised title reads “Molecular Basis for Cdk1 Regulated Timing of Mis18 Complex Assembly and CENP-A Deposition”
2. Reformatted EV Figs 2-5 in portrait mode
3. Added Scale bars to all microscopic images.
4. Added borders to clearly separate gels in Fig EV2B
5. Use of same data in different image panels is explained in the corresponding legends (Fig 4E and 5C).
6. Shortened the manuscript: the total number of characters including spaces is 29,834.

We hope that the revised manuscript is suitable for publication in EMBO reports.

The point-by-point response to the reviewer’s comments is as follows:

Referee #1:

The revised manuscript does an excellent job addressing all of my criticisms of the earlier draft. Their revised model, which is extremely well-supported by their new data, also agrees well with a model reported in eLife by the Musacchio lab after this paper was initially submitted. I think this manuscript should be published essentially as-is. I have a couple notes the authors may wish to address, but I certainly don't need to see the manuscript again.

Notes:

In the section titled "The Mis18alpha MeDiY domain directly interacts with MIS18BP1 20-130 in vitro", the authors should mention the the different behavior of the isolated MeDiY domains of Mis18alpha and Mis18beta is because of alpha's tendency to form homodimers, in contrast to the lack of homodimerization in Mis18beta.

We are glad that this reviewer finds the revised manuscript suitable for publication. The differential ability of Mis18 α _{MeDiY} and Mis18 β _{MeDiY} to oligomerise is mentioned in page 7 of the revised MS.

Figure 3A does not have a calculated molecular weight noted in parentheses. This may or may not have been intentional.

Thanks for pointing out this unintentional mistake. We have now added the calculated MW of Mis18 α/β in Fig 3A.

Referee #2:

Through this revised manuscript, the authors have successfully addressed my key comments and concerns from the previous version of the paper. Based on their changes, and on their explanations to some of my comments, I now find this paper suitable for publication. My one final lingering concern is related to the recent paper from the Musacchio lab (reference 30). There is significant overlap between these papers. I don't think that this is a problem, as two distinct labs establishing this important scientific point definitely helps this field. In addition, this current paper was clearly significantly through the publication process before the publication of the Musacchio lab work. However, I was disappointed by the way that the Musacchio paper was cited, in which the authors essentially added this as a note added in proof, even though they are still able to make substantial changes to their paper. I would much prefer this to be cited more clearly in the discussion, including directly commenting on similarities and potentially any subtle differences between the papers. It may also be good to cite the Musacchio paper at the end of the introduction, or at selected points within the results.

We thank this reviewer for raising this point. Due to space constraints we did not discuss the recently published work from the Musacchio lab in the main text of the previous version. We have now shortened the introduction to accommodate an appropriate discussion in the last paragraph of the revised MS (page 12 and 13).

Referee #3:

In the revised manuscript, the authors have, in parts, attempted to address concerns raised by this reviewer by modifying the text and adding new experimentation (as requested by the other two reviewers). Unfortunately, the authors did not address my two concerns and thus I cannot recommend publication at this time. Please see below for detailed comments:

1) My first point was that the tethering system alone cannot reveal cellular function. The authors rebut by citing many papers that have employed this technology and note in data not shown that the proteins of interest localize to endogenous centromeres. I will note that in the cited papers, the tethering system is meant as a bolstering approach and nearly all of those works probe endogenous proteins at endogenous centromeres at some point in the paper. Thus I feel that these papers in fact bolster my criticism. In the end, my comment remains unchanged; please refer to original review.

2) The authors argue that their work is indeed impactful and mechanistic. I apologize for implying that it was not mechanistic, it is. However I feel that the work adds precision to an already appreciated cellular mechanism. This is subject to opinion and I will leave the editor to decide that. However I also asked that the authors make mutations to endogenous proteins to show that the rates, timing, or levels of CENP-A loading were impacted by these mutations at centromeres. The authors seemingly ignored that comment and therefore I have no recourse but to request it, or at least a reasonable argument, again.

We are disappointed that we failed to convince this reviewer. We stand by our previous argument that alphoid^{neo} array based tethering assay is a powerful and validated tool to dissect structure-function analysis of protein interactions involved in CENP-A deposition in cells (as tethering just Mis18 α is sufficient to recruit other players to facilitate CENP-A deposition at the tethering site). However, we agree with this reviewer that probing endogenous proteins with mutations at endogenous centromeres has the potential to provide insights into their role in influencing the dynamics of CENP-A loading. Unfortunately, these experiments are beyond the scope of this manuscript and due to time constraints we could not perform them at this time.

3rd Editorial Decision

15 March 2017

I am very pleased to accept your manuscript for publication in the next available issue of EMBO reports. Thank you for your contribution to our journal.

YOU MUST COMPLETE ALL CELLS WITH A PINK BACKGROUND

Corresponding Author Name: A. Arockia Jayaprakash

Journal Submitted to: EMBO reports

Manuscript Number: EMBOR-2016-43564V1